



# The high frequency response correction of eddy covariance fluxes. Part 1: an experimental approach for analysing noisy measurements of small fluxes

Toprak Aslan[1], Olli Peltola[2], Andreas Ibrom[3], Eiko Nemitz[4], Üllar Rannik[1], and Ivan Mammarella[1]

[1]Institute for Atmospheric and Earth System Research (INAR)/Physics, Faculty of Science, University of Helsinki, P.O. Box 68, 00014 Helsinki, Finland
[2]Climate Research Programme, Finnish Meteorological Institute, P.O. Box 503, 00101 Helsinki, Finland
[3]Dept. Environmental Engineering, Technical University of Denmark (DTU), Lyngby, Denmark
[4]UK Centre for Ecology and Hydrology (UKCEH), Edinburgh Research Station, Penicuik, Bush Estate, EH26 0QB, UK

**Correspondence:** Toprak Aslan (toprak.aslan@helsinki.fi)

**Abstract.** Fluxes measured with the eddy covariance (EC) technique are subject to flux losses at high frequencies (low-pass filtering). If not properly corrected for, these result in systematically biased ecosystem-atmosphere gas exchange estimates. This loss is corrected using the system's transfer function which can be estimated with either theoretical and experimental approaches. In the experimental approach, commonly used for closed-path EC systems, the low-pass filter transfer function

($H$) can be derived from the comparison of either (i) the measured power spectra of sonic temperature and the target gas mixing ratio or (ii) the cospectra of both entities with vertical wind speed. In this study, we compare the power spectral approach (PSA) and cospectral approach (CSA) in the calculation of $H$ for a range of attenuation levels and signal-to-noise ratios (SNRs). For a systematic analysis, we artificially generate a representative dataset from sonic temperature ($T$) by attenuating it with a first order filter and contaminating it with white noise, resulting in various combinations of time constants and SNRs. For PSA, we

use two methods to account for the noise in the spectra: the first is the one introduced by Ibrom et al. (2007a) ($PSA_{I07}$), where the noise and $H$ are fitted in different frequency ranges and the noise is removed before estimating $H$. The second is a novel approach that uses the full power spectrum to fit both $H$ and noise simultaneously ($PSA_{A20}$). For CSA, we use three different methods: (1) a plain version of Lorentzian equation describing the $H$ ($CSA_H$), (2) a square-root of the $H$ ($CSA_{\sqrt{H}}$), and (3) a square-root of the $H$ with shifted vertical wind velocity time series via cross-covariance maximisation ($CSA_{\sqrt{H},sync}$).

$PSA_{I07}$ tends to overestimate the time constant when low-pass filtering is low, whilst the new $PSA_{A20}$ successfully estimates the expected time constant regardless of the degree of attenuation and SNR. $CSA_H$ underestimates the time constant with decreasing accuracy as attenuation increases due to the omission of the quadrature spectrum. $CSA_{\sqrt{H}}$ overestimates, but its accuracy increases with time-lag correction in the $CSA_{\sqrt{H},sync}$. We further examine the effect of the time constant obtained with the different implementations of PSA and CSA on cumulative fluxes using estimated time constants in frequency response

correction. For our example time series, the fluxes corrected using time constants derived by $PSA_{I07}$ show a bias of $\pm 2\%$. $PSA_{A20}$ showed a similar variation, yet slightly better accuracy. $CSA_H$ underestimated fluxes by up to $4\%$, while $CSA_{\sqrt{H}}$ overestimated them by up to $3\%$, a bias which was mostly eliminated with time-lag correction in the $CSA_{\sqrt{H},sync}$ ($-2\%$ to $1\%$). The accuracies of both PSA and CSA methods were not affected by SNR level, instilling confidence in EC flux





measurements and data processing in setups with low SNR. Overall we show that, when using power spectra for the empirical estimation of parameters of $H$ for closed-path EC systems the new $\text{PSA}_{A20}$ outperforms $\text{PSA}_{I07}$, while when using cospectra the $\text{CSA}_{\sqrt{H},sync}$ approach provides the most accurate results. These findings are independent of the SNR value and attenuation level.

**1 Introduction**

Vertical turbulent fluxes of momentum, energy, and gases between the atmosphere and the biosphere measured by the eddy covariance (EC) technique are subject to both low and high frequency losses (Foken and Napo, 2008; Aubinet et al., 2012). The physical limitations in instrument response times, spatial separation of instruments, line averaging, and air transport through the sampling tubes cause high frequency losses (Aubinet et al., 1999).

The EC sampling system acts as a low-pass filter on the flux and the signal loss must be compensated with the frequency response correction (FRC) during post-processing. The first step in the FRC is the description of the effect of the low-pass filtering of the measurement system, and for this the transfer function approach has been widely used since it was first proposed by Moore (1986). The joint transfer function ($H$) that describes the low-pass filtering of the whole EC system can be determined theoretically or experimentally (Foken and Napo, 2008; Aubinet et al., 2012). The theoretical approach involves a convolution

of various specific transfer functions that are estimated to represent different causes of flux loss. Conversely, in the experimental approach $H$ is estimated from *in situ* measurements. Due to its simplicity, many studies have implemented the theoretical approach, which typically works well with fluxes measured by open-path EC systems as well as momentum fluxes and sensible heat fluxes measured by sonic anemometers (Aubinet et al., 2012). However, for complex EC systems the necessary information to calculate $H$ is not available and needs to be estimated empirically. In addition, the time response of the system can vary

with relative humidity (Ibrom et al., 2007a), tube aging Mammarella et al. (2009), and variations in the flow regime in the tube. Thus, the theoretical approach is not preferred for gas fluxes measured with closed-path EC systems, for which the experimental approach is therefore recommended (Aubinet et al., 2012; Sabbatini et al., 2018; Nemitz et al., 2018).

  In the experimental approach for closed-path systems, $H$ is usually estimated from either the measured power spectra (i.e. PSA) or cospectra (i.e. CSA) of sonic temperature and the mixing ratio of the target gas ($\chi$). Different studies use either PSA

(Ibrom et al., 2007a; Nordbo et al., 2011; Fratini et al., 2012; Sabbatini et al., 2018) or CSA (Aubinet et al., 1999; Humphreys et al., 2005; Mammarella et al., 2009; Peltola et al., 2013). Also, some software packages used for EC flux calculation are based on PSA (e.g. EddyPro, see Biosciences, 2020), while others are based on CSA (e.g. EddyUH, see Mammarella et al., 2016).

  Interestingly, there has not been much debate to date whether to use power spectra or cospectra to determine the time con-

stant of the $H$ (or response time), which characterizes the EC system's high-frequency response. Only recently, Wintjen et al. (2020) investigated the optimal method for high frequency response correction, for fluxes of nitrogen compounds, recommending CSA. Ibrom et al. (2007a) argued for using PSA as the vertical wind speed ($w$) does not contain any relevant information for the spectral attenuation of the gas collection and data acquisition system, allowing to describe sensor related attenuation





independently. This should in principle provides a better estimation of the time constant of the gas analysis system, because the spectral data is not mingled with other components, such as, e.g., sensor separation. In this approach, the effect of sensor separation is then treated explicitly with an additional correction step (Horst and Lenschow, 2009). In most cases the instrumental noise becomes visible in the high frequency range of $\chi'$ power spectra, and this has to be dealt with before the time constant

of the gas sampling system can be estimated. The noise removal procedure is not well established, and this represents a major uncertainty in the PSA approach. This paper explores this uncertainty and proposes a novel, more robust approach to account for the noise in PSA.

This noise is often assumed not to correlate with the fluctuations in $w$, and therefore it does not contribute to the cospectra between $w$ and $\chi$. If this holds, this makes the CSA attractive for the estimation of the time constant, because then the noise

would effectively be disappear from the measured signal. On the other hand, the use of the CSA relies on the correct determination of the time lag between $w$ and $\chi$, which may be difficult in case of small fluxes due to noisier cross-correlation function, making the search for the absolute maximum harder (Langford et al., 2015). Due to the above-mentioned reasons, empirically determined $H$ can be a source of uncertainty for the FRC (Lee et al., 2004). Additionally, the CSA approach inadvertently accounts for the phase shift caused by low-pass filtering, a topic discussed in our companion paper (Peltola et al., 2020).

EC measurements conducted under low flux conditions result in relatively higher noise in signal, i.e. low signal-to-noise ratio (SNR) (Smeets et al., 2009). EC fluxes with low SNR are typically found in many ecosystems especially for methane ($CH_4$) and nitrous oxide ($N_2O$) as well as for other gases and aerosol particles, in which any implementation of the FRC becomes uncertain (Rannik et al., 2015; Nemitz et al., 2018; Oosterwijk et al., 2018). Low SNR can also be observed for $CO_2$ at specific ecosystems (e.g. lakes), for $CH_4$ over well-drained soils or peatlands during winter, for $N_2O$ in long periods

outside the high emission periods (e.g. fertilizer applications, freeze-thaw cycles or rain events), all of which, although small, ultimately contribute to the final flux estimates, and, hence must be corrected from systematic bias. Thus, investigation of uncertainties in commonly used FRC methods is of great importance in order to obtain unbiased, harmonized and continuous time series of gas fluxes measured by EC technique.

To our knowledge, the uncertainty in fluxes caused by the use of the PSA and the CSA have not been investigated system-

atically so far, motivating this study, which hypothesizes that the success of the PSA and CSA usage in FRC depends on the attenuation condition and the level of SNR. Consequently, we expect to see substantially different time constants, correction factors, and eventually different overall magnitudes of correction estimates with respect to the attenuation and SNR conditions. To test this hypothesis, we need a scalar dataset, which represents different attenuation levels and noise conditions. Assuming spectral similarity between scalars, we apply different levels of attenuation and noise to sonic temperature time series ($T$)

in order to generate a proxy representing attenuated gas concentration dataset (e.g. $CH_4$, $N_2O$) with known characteristics. We use a first order low-pass filter, which solely depends on a single time constant ($\tau_{LPF}$), to attenuate the signal. Then, we systematically contaminate the signal with white noise. We then assess which analysis approach most closely retrieves the true time constant used to degrade the flux in the first place. Firstly, we try to retrieve the system time constants using the PSA and CSA, then compare those with original values (i.e., $\tau_{LPF}$). Secondly, in order to demonstrate how variation in time constant

estimation further affects the cumulative fluxes, we low-pass filter a one month $T$ time series, and correct the attenuation with





the FRC of Fratini et al. (2012) via implementing the time constants calculated in the first step. In Sect. 2, the theory of experimental FRC is summarized. In Sect. 3, materials used in this study and methods are explained. Results and discussion are then interpreted in Sect. 4.

## 2 Theory

### 2.1 Background of methods typically used to determine the system time response

In order to calculate the true unattenuated (i.e. frequency-response corrected) covariance ($\overline{w'\chi'}_{\mathrm{corr}}$) with the transfer function method, the measured covariance ($\overline{w'\chi'}_{\mathrm{meas}}$) is multiplied by a correction factor ($F_{corr}$):

$$\overline{w'\chi'}_{\mathrm{corr}} = \overline{w'\chi'}_{\mathrm{meas}} F_{corr}. \tag{1}$$

One way to calculate $F_{corr}$ is to estimate the ratio of the integrated unbiased and biased cospectra as a function of frequency. In order to define the co-spectrum of the unattenuated scalar under the assumption that the normalized cospectrum of all scalars has the same form (i.e scalar similarity), either the cospectra model (see Mammarella et al., 2009) or the measured cospectra (see Fratini et al., 2012) of $T$ are used as a reference. Many studies used the surface layer models described by Kaimal et al. (1972) and based on Kansas experiments (see Moore, 1986). The attenuated cospectra are obtained by convolution of the reference cospectra with the transfer function ($H$), which characterizes the filtering of the EC system.

Another way to calculate $F_{corr}$ is to simulate the attenuation with a recursive filter in time, rather than in frequency space, in which $F_{corr}$ is defined as the ratio of the unattenuated and attenuated co-variances (see Goulden et al., 1997). In addition, based on the same approach, an experimental method was proposed by Ibrom et al. (2007a) to parameterise the correction factor separately for stable and unstable stratifications, using meteorological data. For this method, a further modification was later suggested by Fratini et al. (2012) for the processing of large fluxes.

Regardless of the method chosen, $H$ needs to be obtained either theoretically or empirically before $F_{corr}$ can be calculated. In empirical approach, $H$ can be determined using *in situ* measurements as a ratio of the normalized power spectra (for PSA) or cospectra (for CSA) of the attenuated scalar to those of an unattenuated scalar, e.g., $T$. In both approaches, in order to reduce the uncertainties on the low frequency part of the spectra, and to fulfill the assumption of spectral similarity, data must be selected from periods with rigorous stationary turbulent mixing. In addition, the power spectra and cospectra of $T$ and $\chi$ should be normalized with their standard deviations so that they can be compared with each other (see their Eq. 2 in Ibrom et al., 2007a).

For the PSA, $H$ can be calculated using the power spectra of $\chi$ and $T$ (Eq. (2)), where the effect of sensor separation should additionally be treated via the method proposed by Horst and Lenschow (2009). For PSA, $H$ is derived as:

$$H_{PSA}(f) = \left[\frac{S_\chi(f)}{\sigma_\chi^2}\right]\left[\frac{S_T(f)}{\sigma_T^2}\right]^{-1}, \tag{2}$$

where $S_\chi$ indicates the power spectrum of measured target gas mixing ratio, $S_T$ is the power spectrum of $T$ and variances ($\sigma_\chi^2$ and $\sigma_T^2$) are calculated across the frequency range over which no attenuation occurs. Instrumental noise often becomes





dominant in the high frequency range of the power spectrum and also contributes to $\sigma_s$. Where instrumental noise is large, the power spectrum asymptotically approaches a line of slope +1 at the high frequency end (Fig. 1) which represents the fact that white noise affects all frequencies equally, but is multiplied by $f$ in this representation (see Sect. 2.2). Thus, prior to calculation of Eq. (2), the noise contribution to the power spectra of $S_\chi$ should be removed and this is typically done by fitting a line to the
high frequency end (Ibrom et al., 2007a). Finally, the frequency dependence of $H_{PSA}$ can be be described through a sigmoidal curve which is characterised by the time constant ($\tau$) of the measurement system (see for more details Peltola et al., 2020, and references therein):

$$H_{emp}(f) = \frac{1}{1 + (2\pi f \tau)^2}. \tag{3}$$

For PSA, Ibrom et al. (2007a) updated Eq. (2) by introducing a normalisation factor, which is used to secure the spectral
similarity especially for small fluxes. As described in their Eq. (6), the time constant and the normalisation factor ($F_n$) are obtained via fitting the following equation to the dampened and noise free $\chi$ data:

$$\frac{S_\chi(f)}{\sigma_\chi^2} = \frac{S_T(f)}{\sigma_T^2} F_n \frac{1}{1 + (2\pi f \tau)^2}. \tag{4}$$

In our study, we follow the same procedure (hereafter $PSA_{I07}$) for the time constant calculation for the PSA, which is summarized in Sect. 3.2. In addition to $PSA_{I07}$, we used a new comprehensive method for PSA, which is summarized in Sect.
15  2.2.

Alternatively, for the CSA, $H$ is calculated as

$$H_{CSA}(f) = \left[ \frac{Co_{w\chi}(f)}{\overline{w'\chi'}} \right] \left[ \frac{Co_{wT}(f)}{\overline{w'T'}} \right]^{-1}, \tag{5}$$

where $f$ is the natural frequency, $Co_{w\chi}$ indicates the cospectrum of measured $w$ and target gas mixing ratio $\chi$; $Co_{wT}$ is the cospectrum of measured kinematic heat flux, $\overline{w'\chi'}$ and $\overline{w'T'}$ are covariances calculated across a frequency range where the
cospectra are not attenuated.

For CSA, $\tau$ is obtained similar to PSA via fitting $H_{emp}$ to Eq. (5), however, there is an ongoing debate on whether the correct transfer function for the cospectra would be $\sqrt{H_{emp}}$, instead of $H_{emp}$ (Moore, 1986; Eugster and Senn, 1995; Horst, 1997, 2000; Fratini et al., 2012; Hunt et al., 2016) as discussed throughouly by Peltola et al. (2020) who also provide a mathematical derivation on the subject. Since both forms exist widely in the literature, here we opt to apply both approaches to CSA to
demonstrate discrepancies between the approaches.

## 2.2 A new approach to estimate the time constant from a noise-contaminated power spectrum

In the $PSA_{I07}$ application to noisy data, the time constant is typically obtained via two separate fitting procedure steps following the approach of Ibrom et al. (2007a) as illustrated in Fig. 1 for a noisy spectrum. First the noise part is fitted with a line of an unconstrained slope (blue dashed line) and then the transfer function is fitted (black solid line). This approach can be
problematic as we will show in Sec. 4.1 below, because if the resulting line has a slope less than unity, its extrapolation to lower frequencies erroneously removes true signal. By contrast, a line with a predefined slope of 1 can only be fit for very


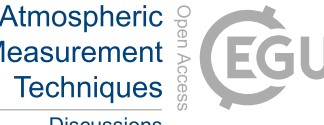

noisy spectra where the 1:1 is fully reached. Here we introduce a new alternative approach (hereafter $\mathrm{PSA}_{A20}$), which performs the estimation of the time constant and accounts for instrument noise in a single non-linear comprehensive fitting step.

Equation (4) can be extended to the following equation, which includes also the noise component of the scalar:

$$f\frac{S_\chi(f)}{\sigma_\chi^2} = f\frac{S_T(f)}{\sigma_T^2}F_n\frac{1}{1+(2\pi f\tau)^2} + f\frac{S_{\chi,n}(f)}{\sigma_\chi^2}, \tag{6}$$

5     where $S_{\chi,n}(f)$ is the power spectrum of the noise in $\chi$. Here it is assumed that the noise and signal in measured $\chi$ time series are uncorrelated and hence two independent and additive components of the time series. In the case of white noise, Eq.( 6) can be simplified to

$$f\frac{S_\chi(f)}{\sigma_\chi^2} = f\frac{S_T(f)}{\sigma_T^2}F_n\frac{1}{1+(2\pi f\tau)^2} + fb, \tag{7}$$

where $b$ is the y-axis intercept of the power spectra of the white noise multiplied with $f$, which is shown linearly in log-log

10     scale. All terms in the equation have been multiplied by $f$ because this is the standard normalisation used to depict the spectral density functions (Fig. 1). The detailed derivation of Eq. (7) can be found in Appendix A.

In this approach, the frequency ranges used for fitting both for transfer function and noise are not separated, hence the fitting is performed across the entire frequency domain.

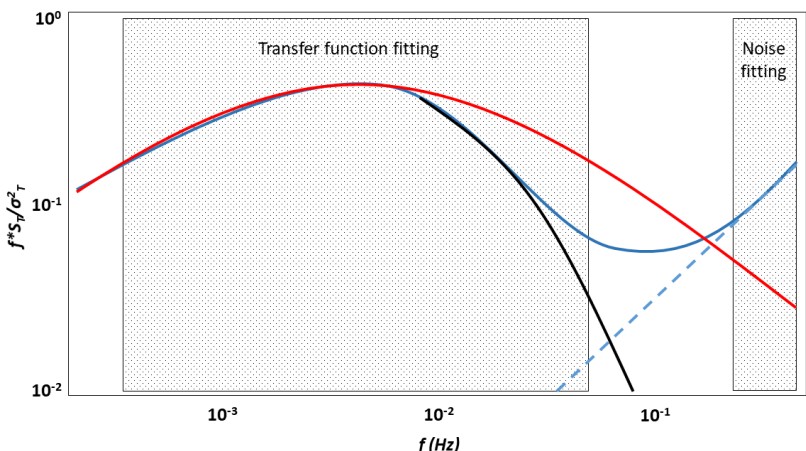

**Figure 1.** A diagram illustrating two-steps fitting procedures (i.e. noise removal and transfer function) to estimate time constant for $\mathrm{PSA}_{I07}$. Shown are the spectra of unattenuated and noise-free scalar (red line), and spectra of low-pass filtered and noisy scalar (dark blue) and after (black) noise removal. The blue-dotted lines are the curves fitted to the high frequency end of blue line, then extended towards lower frequencies. The shaded areas mark the frequency range used for fitting for noise removal and transfer function.





## 3 Materials and methods

### 3.1 Sites description and measurements

Two datasets measured with sonic anemometer in an EC set up from the Siikaneva fen site were used in this study. The site is located in Southern Finland ($61^o49.9610'$ N, $24^o11.5670'$ E; 160 m a.s.l). The data were measured with 10 Hz sampling

frequency using a 3-D sonic anemometer (Model USA-1; Metek GmbH, Elmshorn, Germany). Further details about the site and measurements can be found in Peltola et al. (2013).

The first dataset ($D_1$) was used for the time constant calculation (see Sect. 3.2). It contained 70 half-hourly EC data records measured in fully turbulent daytime conditions in the period from May to September 2013 with an average sensible heat flux of $114.3\,\mathrm{W\,m^{-2}}$, friction velocity of $0.3\,\mathrm{m\,s^{-1}}$, and wind speed of $2.1\,\mathrm{m\,s^{-1}}$.

The second dataset ($D_2$) was used for the cumulative flux calculation based on $T$ (see Sect. 3.3). This dataset was measured between May 1 and May 31 2013 and consisted of 1440 half-hourly periods for which fluxes were calculated.

### 3.2 Data processing for time constant estimation

The data processing flow for all variants of PSA and CSA is summarized in Fig. 2. In order to generate the artificial dataset, which represents various known levels of SNR and attenuation, we first applied commonly used EC data processing procedures,

i.e., de-spiking, two-dimensional coordinate rotation of the wind velocity vector, and linear de-trending to $D_1$. We did not apply the time lag correction because we assumed that there was no time-lag between $T$ and $w$, both of which measured with the same instrument. We then degraded each half-hourly $T$ time series with a first order low-pass filter in the spectral domain (see Sect. 3.2.1) and contaminated it with prescribed amount of white noise in the time domain (see Sect. 3.2.2).

For PSA, we calculated power spectra of $T$, following Sabbatini et al. (2018), normalised it by the total variance calculated

within the frequency range of 0.0012 and 0.05 Hz, and averaged it into a logarithmically equally distanced frequency base. Then we took an ensemble average of the 70 power spectra. For $\mathrm{PSA}_{I07}$, we first removed the noise from the power spectra (see Sect. 3.2.3), then retrieve the time constant (i.e. $\tau_{PSA_{I07}}$) via fitting Eq. (4) within a frequency range, the lower limit of which is 0.01 Hz, while the optimal higher limit is defined via visual inspection. For $\mathrm{PSA}_{A20}$, we obtained the time constant (i.e. $\tau_{PSA_{A20}}$) via fitting Eq. (4) using the entire frequency domain.

Low-pass filtering introduces a time-lag in the scalar of interest in addition to any physical time-lag that may be caused by transport through sampling lines and/or sensor separation (Massman, 2000; Ibrom et al., 2007b). In our case, since the scalar of interest is represented by the attenuated $T$, this time-lag occurs between $T$ and $w$, two entities that are not separated by a further physical time-lag. In order to demonstrate the effect of low-pass induced time-lag on the time constant estimation in CSA, we generated a dataset in which the time-lag was adjusted via the maximisation of the crosscovariance, which was

explored in addition to the original not time-lag adjusted dataset. Next, we calculated cospectra of $T$ and $w$ of both datasets, normalised it with the total covariances calculated within the frequency range of 0.0012 and 0.05 Hz, and averaged the data into exponentially spaced frequency bins. Then we calculated ensemble averages. Lastly, we derived the time constants for the original dataset (i.e. $\tau_{CSA_{\sqrt{H},sync}}$) via fitting the square-root of Eq. (3), while for time-lag corrected dataset, i.e. $\tau_{CSA_H}$ via





fitting Eq. (3), and (i.e. $\tau_{CSA_{\sqrt{H}}}$) via fitting square-root of Eq. (3). All fits were estimated for the frequency range from 0.01 to 2 Hz. For a more complete analysis on the effect of low-pass induced time-lag on time constant estimation see the companion paper (Peltola et al., 2020).

In summary, for the two methods of PSA and three methods of CSA, we assessed 45 different conditions each, combining five
different attenuation levels with nine different SNRs. We repeated the same procedure 100 times to account for the uncertainty associated with the white noise generation, and thus obtained 100 different values for the time constant for all attenuation and SNR levels for both PSA and CSA. The relevant results are shown in Sect. 4.2.

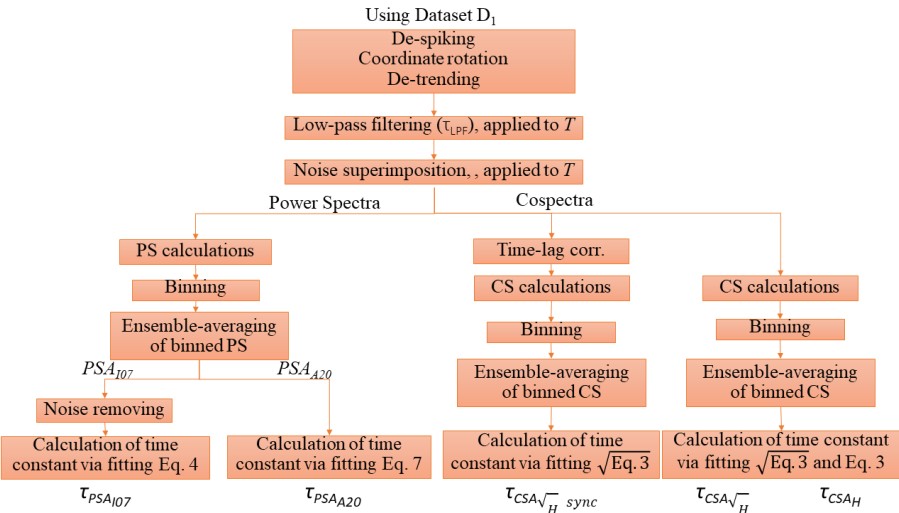

**Figure 2.** Flow chart of the data processing for time constant calculation using the dataset $D_1$. $\tau_{LPF}$ is the time constant of the first order filter. $\tau_{PSA_{I07}}$ and $\tau_{PSA_{A20}}$ represent the estimated time constants with power spectra (PSA), while $\tau_{CSA_{\sqrt{H},sync}}$, $\tau_{CSA_{\sqrt{H}}}$ and $\tau_{CSA_H}$ with cospectra (CSA) approaches. $H$ is the spectral transfer function.

### 3.2.1 Low-pass filtering

The dynamic performance of any EC measurement system can be approximated with a linear first-order non-homogeneous
ordinary differential equation (Massman and Lee, 2002):

$$\tau_{LPF}\frac{d\chi_O}{dt} + \chi_O(t) = \chi_I(t), \tag{8}$$

where $\chi_O$ is the output of a scalar sensor, $\chi_I(t)$ is the true scalar concentration (i.e., input) and $\tau_{LPF}$ is the characteristic time constant of the sensor response (Massman and Lee, 2002). The spectral response ($h_{LPF}(\omega)$) of such system can be obtained by the Fourier transform of the ratio of output signal to the input signal, i.e. $\chi_O/\chi_I$ (Horst, 1997):

$$h_{LPF}(\omega) = \frac{1}{1 - j\omega\tau_{LPF}}, \tag{9}$$





where $\omega = 2\pi f$ and $j = \sqrt{-1}$.

The desired (i.e. low-pass filtered) output data in frequency domain can be estimated as

$$Z_O = h_{LPF}(\omega)Z_I. \tag{10}$$

where $Z_I$ is the Fourier transform of $\chi_I$. From this the low-pass filtered time series $\chi_O$ can be acquired by applying the inverse
Fourier transform to $Z_O$ and taking the real part. In practice, the complex conjugate of $h_{LPF}$ is used to derive the correct
temporal lag (scalar lag with respect to $w$).

We followed this procedure to filter $T$ with five different $\tau_{LPF}$ values, i.e. 0.1, 0.2, 0.3, 0.4 and 0.5 s, corresponding to $f_c$
values of  1.60, 0.80, 0.53, 0.40 and 0.32 Hz, respectively.

### 3.2.2  Noise superimposition

In this study we use Gaussian white noise, which has equally distributed spectral densities across all frequencies (Stull, 2012),
to contaminate the filtered signal in time space. In order to generate time series with varying levels of SNR, we first generated
white noise with unit standard deviation and multiplied the white noise with the standard deviation of the original $T$ time series
with different ratios (e.g. from 0.1 to 0.9). This represents the amount of noise compared to the amount of signal (e.g. from 10
to 90%). We then added this noise to the filtered signal. As a result, we obtained SNR values of 10.0, 5.0, 3.3, 2.5, 2.0, 1.6, 1.4,
1.2, and 1.1, which are equal to the ratio of the standard deviation of $T$ and the standard deviation of white noise.

### 3.2.3  Noise removal

The instrumental noise is often removed via the approach introduced by (Ibrom et al., 2007a), in which, to detect the noise, a
line with unconstrained slope is fitted to the power spectra in logarithmic frequency space at a certain frequency range, which is
assumed to be solely dominated by noise. Then, the fitted line is extrapolated towards lower frequencies and finally subtracted
from the original spectra in logarithmic frequency space as described in Ibrom et al. (2007a) (cf. dashed lines in Fig. 4). The
boundaries of the frequency domain of fitting are defined by visual inspection of the power spectra. In our study, the optimal
frequency domains used for noise fitting were detected as 3, 2.6 and 2.3 to 5 Hz for the different attenuation level of 0.1, 0.2
and 0.3 s, respectively. For attenuation levels of 0.4 and 0.5 s, we used the frequency range of 2 to 5 Hz.

### 3.3  Data processing when estimating long-term budgets

Data processing steps when estimating the long-term budgets using the $D_2$ dataset are summarized in Fig. 3.

We applied regular EC data processing, which included de-spiking, coordinate rotation, de-trending. Next, the $T$ time series
were deteriorated with values of $\tau$ that varied between 0.1 and 0.5 s to mimic a realistic range of scalar attenuations (mimicking
the conditions, e.g., of $CH_4$, $N_2O$). Later, the low-pass induced time-lag was accounted for via maximisation of the cross-





covariance, which was followed by the calculation of the covariances. Then we applied the frequency response correction using Eq. (1), where $F_{corr}$ was estimated using the method proposed by Fratini et al. (2012)[1]:

$$
F_{corr} = \frac{\int\limits_{f=f_{min}}^{f_{max}} CO(f)df}{\int\limits_{f=f_{min}}^{f_{max}} CO(f)\sqrt{H_{emp}(f)}df}, \tag{11}
$$

where $CO$ equals the current $T$ cospectrum when the absolute sensible heat fluxes exceeded 15 W m$^{-2}$. For small fluxes we
used a site specific cospectral model (see Appendix D) for $CO$ instead of parameterisation of $F_{corr}$ proposed by Ibrom et al.
(2007a). To make sure that the analysis was not affected by low data quality, we removed the fluxes with low friction velocity
($u_* < 0.2 \ ms^{-1}$), unrealistic sensible heat fluxes and non-stationary conditions (Foken and Wichura, 1996), eliminating 553
half-hourly data points out of 1440 (ca. 38%).

The time constant ($\tau$) in $H_{emp}$ is either estimated by PSA (i.e., $\tau_{PSA_{I07}}$ and $\tau_{PSA_{A20}}$) or CSA (i.e., $\tau_{CSA_{\sqrt{H},sync}}$, $\tau_{CSA_{\sqrt{H}}}$
and $\tau_{CSA_H}$), and this yielded cumulative fluxes $F_{PSA_{I07}}$, $F_{PSA_{A20}}$, $F_{CSA_{\sqrt{H},sync}}$, $F_{CSA_{\sqrt{H}}}$ and $F_{CSA_H}$, respectively. For sim-
plicity, in the FRC only median values of time constants of PSA and CSA ensembles estimated were used for each combination
of SNR and attenuation. The reference fluxes ($F_{REF}$) were estimated with $F_{corr}$ calculated as the ratio of unattenuated and
attenuated covariances $\overline{T'w'}$, and maximisation of cross-covariance was used to account for the time-lag induced by the low-
pass filter. We present the differences between cumulative fluxes as relative differences with respect to the reference ($F_{REF}$) in
%. In particular, as an example, the relative bias for $PSA_{I07}$ is calculated as $100(F_{PSA_{I07}} - F_{REF})/F_{REF}$, while for $CSA_H$
as $100(F_{CSA_H} - F_{REF})/F_{REF}$ for several attenuation and SNR conditions. The relevant results are shown in Sect. 4.3.

## 4   Results and Discussion

### 4.1   Time constant estimation with PSA

In order to provide an illustration of the important steps in the data analysis in PSA (e.g. low-pass filtering, noise superimposi-
tion and noise removal only for $PSA_{I07}$), here we show how the shape of power spectra changes on a logarithmic scale for a
few SNR values (5, 3.3 and 2.5) and low-pass filtering conditions (Fig. 4). The low-pass filter time-constants ($\tau_{LPF}$) of 0.1, 0.3
and 0.5 s result in $f_c$ values beyond which the signals become attenuated of ca. 1.6, 0.5 and 0.3 Hz. As time constant increases,
$f_c$ decreases, causing stronger attenuation. Referring to the upper panel of Fig. 4, for constant attenuation ($\tau_{LPF} = 0.1$ s), if
the SNR decreases, noise becomes more visible and the line fit to the noise becomes more consistent. Similarly, referring to
the left panel of Fig. 4, for constant SNR (e.g. for SNR=5), from top to bottom, as attenuation increases, the slope of the fit
increases and therefore, goodness of fit increases (Table 1). As discussed in Sect. 2.2, white noise causes a slope of one, and
thus any discrepancies from one results in overestimated noise removal. Based on Fig. 4 alone, it is evident that removing

---

[1]Here we preferred using the square-root to describe the true transfer function as it is a good approximation when maximisation of the cross-covariance is
used for the time-lag correction as shown by Peltola et al. (2020).





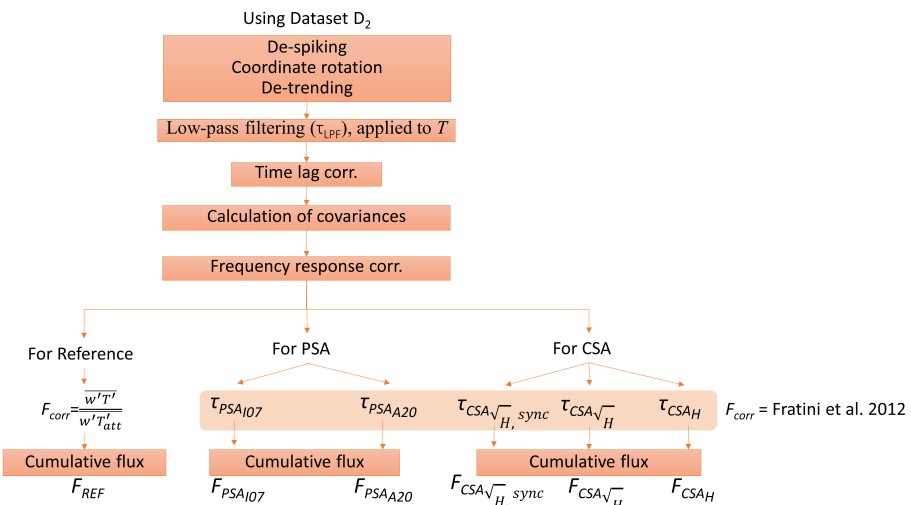

**Figure 3.** Flow chart of the data processing for the cumulative flux calculation using $T$ of dataset D$_2$. $F_{REF}$ refers to reference cumulative flux obtained via frequency response correction applied, where the correction factor ($F_{corr}$) is calculated as the ratio of covariances of unattenuated and attenuated of $T$ and $w$. $F_{PSA_{I07}}$, $F_{PSA_{A20}}$, $F_{CSA_{\sqrt{H},sync}}$, $F_{CSA_{\sqrt{H}}}$ and $F_{CSA_H}$ are the cumulative fluxes corrected via following Fratini et al. (2012), where $F_{corr}$ is calculated via implementing the time constants (for PSA; $\tau_{PSA_{I07}}$ and $\tau_{PSA_{A20}}$, for CSA; $\tau_{CSA_{\sqrt{H},sync}}$, $\tau_{CSA_{\sqrt{H}}}$ and $\tau_{CSA_H}$) calculated in previous section.

**Table 1.** Results of the noise removal procedure applied in the power spectra approach following Ibrom et al. (2007a) ($PSA_{I07}$): the values indicate the slopes of fitted line to the high frequency end of the spectrum in Fig. 4, with the coefficient of determination ($R^2$) shown in the parenthesis, as a function of $\tau_{LPF}$ and SNR. Note that for accurate noise removal the slopes should equal one.

|  | SNR=5 | SNR=3.3 | SNR=2.5 |
|---|---|---|---|
| $\tau_{LPF}$=0.1 s | 0.57 (0.93) | 0.78 (0.98) | 0.87 (0.99) |
| $\tau_{LPF}$=0.3 s | 0.91 (0.99) | 0.96 (0.99) | 0.97 (0.99) |
| $\tau_{LPF}$=0.5 s | 0.96 (0.99) | 0.98 (0.99) | 0.99 (0.99) |

the noise from power spectra can be done with higher accuracy when the high frequency attenuation increases and/or SNR decreases.

The results of the time constant estimation are shown in Fig. 5a, and 5b for PSA$_{I07}$ and PSA$_{A20}$, respectively. Results are presented as medians with $25^{th}$ and $75^{th}$ percentile ranges for the repeat simulations. It can be seen that for PSA$_{I07}$ the interquartile range (IQR) expands as the amount of noise increases, while for PSA$_{A20}$ it is small and almost constant with only a slight increase.

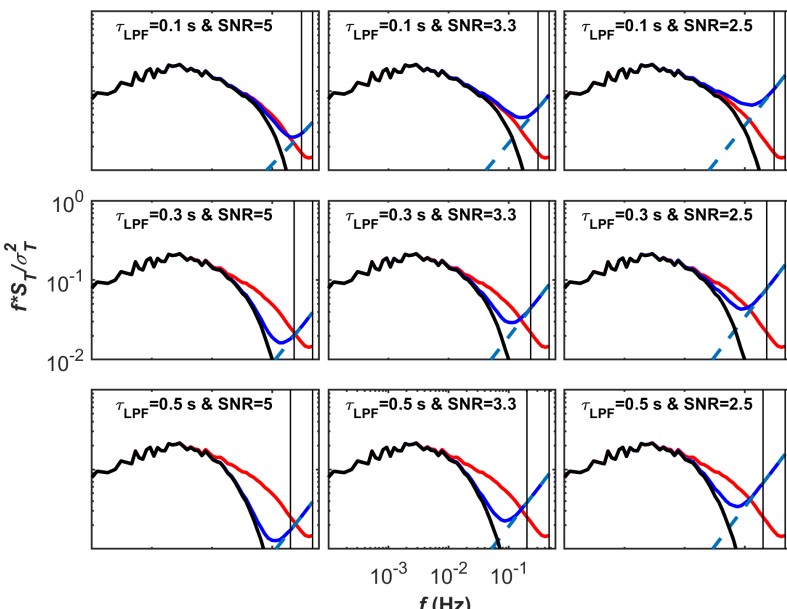

**Figure 4.** Effect of several low-pass filtering ($\tau_{LPF}$ = 0.1, 0.3, and 0.5 s) and SNR (5, 3.3 and 2.5) on spectra of sonic temperature ($T$) of 70 half-hourly data illustrated in logarithmic scale, where $f$ is natural frequency, $S_T$ is spectral density, $\sigma_T^2$ is variance of $T$. Shown are the spectra of raw measured sonic temperature (red lines), and of the artificially deteriorated (i.e. low-pass filtered and noise superimposed) sonic temperature before (dark blue) and after (black) noise removal following Ibrom et al. (2007a), through subtraction of the linear fit (blue dashed lines) to the high frequency end of the deteriorated spectra. The vertical lines mark the frequency range used for fitting for the noise removal. The lower thresholds of the frequency range are 3, 2.3 and 2 Hz for the attenuation levels of 0.1, 0.3 and 0.5 s, respectively.

PSA$_{I07}$ overestimates the time constants with improving accuracy from low attenuation to high attenuation conditions regardless of the SNR values. The overestimation is likely due to the noise removal procedure, which further attenuates the high-frequency end of the spectra via removing the part of the signal together with noise (i.e. the noise is fitted with a slope < 1). In this approach, the accuracy of the noise removal procedure can be improved by visual inspection and adjustment of the fitting range to provide slopes close to one, and thus better fitting parameters. However, especially for low attenuation conditions (e.g. 0.1, 0.2 s), the frequency range is not sufficient to detect the noise statistically, meaning that the linear fitting method is not ideal for differentiating in the spectral power the noise contribution from the real variations due to turbulence. In addition to shortcoming of the linear fitting, the visual inspection of both detecting the frequency ranges for noise removal and the $H$ fitting constitutes another uncertainty source due to its subjectivity. It requires expertise on the topic, and is hard to

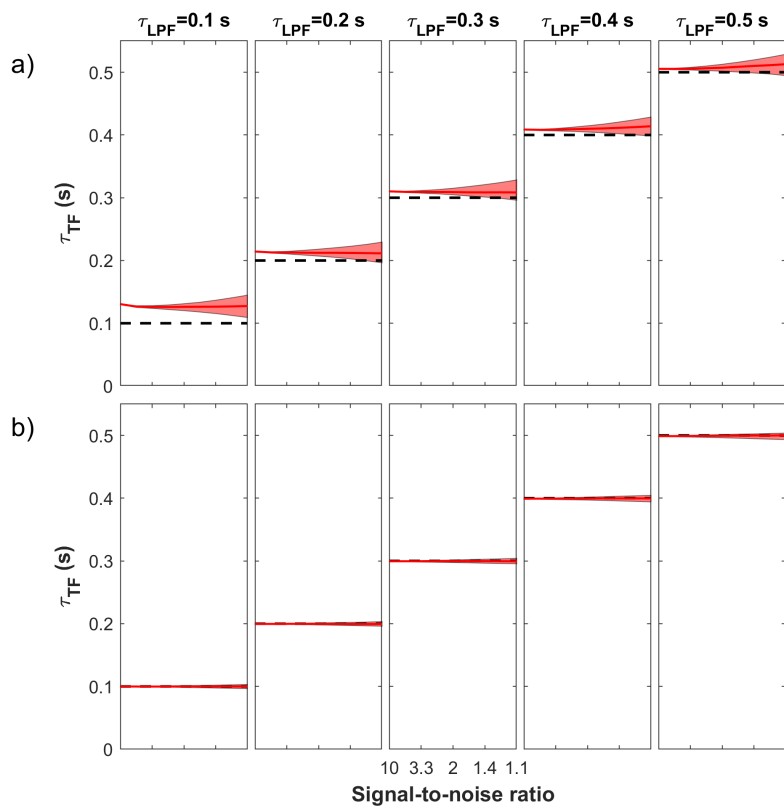

**Figure 5.** Time constants calculated using the power spectra approach, comparing (a) $PSA_{I07}$, and (b) $PSA_{A20}$, in several low-pass filtering condition ($\tau_{LPF}$=0.1 - 0.5 s) as a function of signal-to-noise ratio (SNR) over the range 10-1.1, which correspond to the amount of noise (e.g. 10-90 %). The solid red curve represents the median, while the shaded area represent inter-quartile ranges. The dashed line corresponds to the expected value ($\tau_{LPF}$), which was used for the artificial low-pass filtering.

automise when using software used for flux calculations (e.g., EddyPro). Moreover, in some cases the optimal visual inspection might not be sufficient as the low attenuation results in our study suggest. In our case we could not further improve the accuracy even though the exact attenuation and SNR level were known, which is not the case with real world data.

$PSA_{A20}$ successfully estimates the time constants regardless of the attenuation and SNR level. This is due to using the whole
5    frequency range for fitting without separating the superimposed attenuated signal and noise. The most important advantage of the $PSA_{A20}$ is that it does not require visual inspection.

We assumed that the noise contaminating the signal is white noise, which may not always be the case in real world data. Thus, the accuracy of the $PSA_{A20}$ depends on knowledge of the nature of the noise, which should be determined in advance,





and implemented in Eq. (6) by adjusting the last term that characterises the noise. This said, in many EC studies, the type of the noise is attributed to white noise (e.g., Launiainen et al., 2005; Peltola et al., 2014; Rannik et al., 2015; Gerdel et al., 2017; Wintjen et al., 2020)[2], but brown noise in other studies (Wintjen et al., 2020). A simple approach to characterise the type of noise is either by examining the high frequency end of spectra, which is similar to our study, or through the Allan variance, e.g.

Werle et al. (1993). Alternatively, we conducted a brief investigation into the type of noise by comparing the power spectra of measurements with very low SNR and those of white and blue noise (see Appendix C). It should be noted, however, that there are situations when noise is complex and difficult to predict $a\ priori$.

## 4.2 Time constant estimation with CSA

Figure 6 illustrates cospectra of three low-pass filtered cases (i.e. $\tau$=0.1, 0.3 and 0.5 s) with most noisy conditions (i.e. lowest
SNR of 1.1) in addition to raw (i.e. original) cospectra. It shows that noise contamination does not affect the shape of the cospectra.

$CSA_H$ systematically underestimates the time constant, while, in contrast, $CSA_{\sqrt{H}}$ overestimates, the accuracy of which is further significantly improved when the time-lag correction is applied to the time-series, in $CSA_{\sqrt{H},sync}$ (Fig. 7). The reason for this improvement is discussed in detail in the companion paper (Peltola et al., 2020).

The accuracy of $CSA_H$ decreases with increasing attenuation. The spectral energy of two variables comprises a real part, i.e. the cospectrum ($Co$), and an imaginary part, i.e. the quadrature spectrum ($Q$) (Stull, 2012). The latter is often neglected by assuming the attenuation does not change the phase between the two variables (Horst, 2000). In order to test this assumption we further investigated the effect of $Q$ on the time constant estimation (see Appendix B). In short, application to a couple of case studies shows that $CSA_H$ can under- or over-estimate the time constant due to neglected $Q$, depending on the sign of $Q$.

The bias is larger for higher attenuation since the central term ($2\tau_c Q/Co$ term in Eq. (B1)) increases linearly with $\tau$. We showed that (Table B1 and B2), the bias can be eliminated when the ratio of $Q$ over $Co$, both of which are obtained from unattenuated records, is utilized in the $H$ (Eq. (B1)). However, the use of the ratio is not applicable in real-world data of closed-path systems where $Q$ and $Co$ are distorted in case of attenuated time series, hence this uncertainty can be noted as a shortcoming of the CSA.

In the case of $CSA_{\sqrt{H},sync}$, the variation around the expected values can be attributed to the shortcomings in the maximisation of the crosscovariance used for the time-lag correction, the precision of which is limited to the sampling interval (e.g. 0.1 s). Hence, the importance of the precision of time-lag detection can be noted as a shortcoming of CSA.

In practice, the time-lag correction when utilized with the square-root of $H$ can eliminate the above mentioned phase effect, the theoretical premise of which is further investigated in Peltola et al. (2020).

Lastly, as expected, the level of noise does not affect the accuracy of any of the CSA methods at any SNR level as the random noise does not correlate with $w$, which is a pivotal advantage of the CSA in general. It does, however, add random noise to the result that is greater than in the improved $PSA_{A20}$ approach.

---

[2]This also includes Ibrom et al. (2007a), who falsely interpreted the (white) noise as blue. This conclusion, however, was misguided by neglecting the fact that the power spectra were multiplied with $f$.



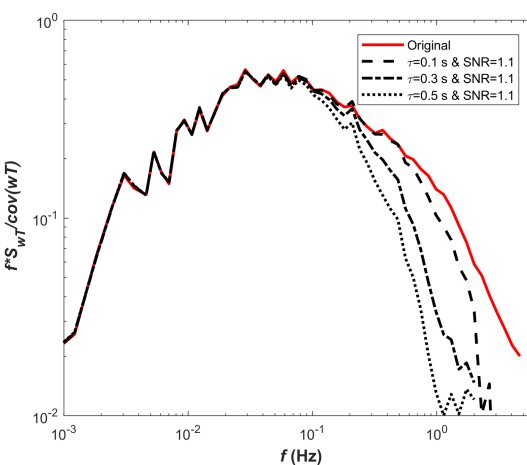

**Figure 6.** Normalised ensemble cospectra for the original and various attenuated time series (i.e. $\tau$=0.1, 0.3 and 0.5 s). White noise was added at a signal to noise ratio (SNR) of 1.1. The red line represents the non-attenuated and noise free cospectrum, while the dashed, dotted-dashed and dotted lines show the noise contaminated and low-pass filtered cospectra for the three attenuation levels.

## 4.3 Effect of time constant variations on cumulative fluxes

Figures 8 and 9 illustrates how variation in the estimation method of the time constant affected the cumulative fluxes in comparison to reference fluxes (see Sect.3.3) calculated using dataset $D_2$.

$PSA_{I07}$ based fluxes showed a bias of $\pm2\%$ with negligible response to the SNR level, whereas $PSA_{A20}$ showed a some-
what similar behaviour with slightly better accuracy reflecting the more accurate time constant estimation previously shown. Fluxes based on $CSA_{\sqrt{H},sync}$ were very close to the expected value with similar biases as the PSA methods. $CSA_{\sqrt{H}}$ overestimates the fluxes up to 3%, while the $CSA_H$ method underestimates fluxes by up to 4%. These findings are partly consistent with the observed biases on time constant estimation, meaning that where the time constant and the low-pass filtering were overestimated (e.g. with the $PSA_{I07}$ especially for $\tau$=0.1 s and $CSA_{\sqrt{H}}$), the spectral correction factor and thus the fluxes were overestimated, too. In summary, the findings indicate that using PSA methods (particularly $PSA_{A20}$) for time constant estimation provides reasonably accurate FRC. On the other hand, for CSA methods, $CSA_{\sqrt{H},sync}$ better approximates the time constants, hence FRC. These results are in agreement with the analysis presented in Peltola et al. (2020), showing that $CSA_{\sqrt{H},sync}$ well approximates the effect of phase shift on the estimation of the time constant and flux correction factor. See more details in Peltola et al. (2020).

## 4.4 Review of typical signal-to-noise ratios and response times encountered during closed-path flux measurements

The range of attenuation and SNR conditions reported in the literature is rather wide and varies depending on ecosystem type, the scalar of interest, data processing, configuration of instruments, and setup of EC system. Ibrom et al. (2007a) examined





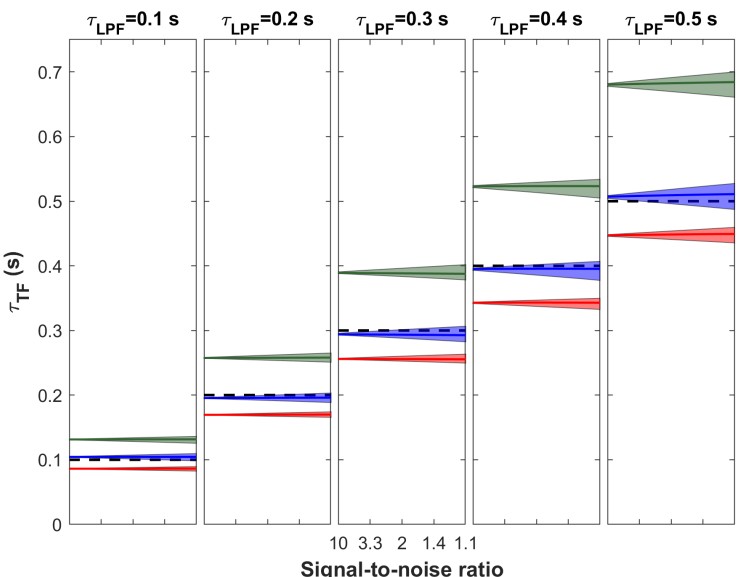

**Figure 7.** Time constants calculated using the cospectral approaches, i.e. $CSA_H$ (red), $CSA_{\sqrt{H}}$ (green), and $CSA_{\sqrt{H},sync}$ (blue), for several low-pass filtering condition ($\tau_{LPF}$=0.1 - 0.5 s) as a function of signal-to-noise ratio (SNR) over the range 10-1.1, which correspond to the amount of noise (e.g. 10-90 %). The solid curves represent the medians, while the shaded areas represent iner-quartile ranges. The dashed line corresponds to the expected value ($\tau_{LPF}$), which was used for the artificial low-pass filtering.

fluxes of water vapour and $CO_2$ measured over a temperate forest, both of which were disturbed by noise in the high frequency range of power spectra (see their Fig. 2). They identified an $f_c$ of $CO_2$ as low as 0.325 Hz (i.e. ca. $\tau = 0.5$ s), indicating strong attenuation, while $H_2O$ showed even stronger attenuation which increased with relative humidity to up to 0.010 Hz (i.e. ca. $\tau = 16$ s) due to the sorption/ desorption effects on the sampling line internal walls. Langford et al. (2015) reviewed

SNR values of various gases published in many studies. In their comprehensive analysis (their Fig. 4), the majority of the half-hourly datasets of isoprene and acetone, measured with quadrupole-based Proton Transfer Reaction-Mass Spectrometer (PTR-MS, Ionicon Analytik GmbH, Austria) above broad leaf woodland, roughly showed SNRs of 0.5, 0.3, respectively, and benzene measured (with same instrument) in the urban environment showed a SNR of 0.3. Additionally, $N_2O$ measured with an older tunable diode laser (Aerodyne Research Inc., Billerica, MA, USA) over managed grassland showed a SNR of one.

Rannik et al. (2016) examined the random uncertainties in fluxes measured over forest, lake and peatland ecosystems in the Boreal region. According to their unpublished calculations, the SNRs of $CO_2$ and $H_2O$ measured with an infrared gas analyzer (Li-6262, LICOR, USA) were 2.6 and 10.3 for the forest site respectively, while the SNRs of $CO_2$, $H_2O$ and $CH_4$ measured with two closed-path analysers (i.e. LI-7000, LI-COR Inc., Lincoln, NE, USA for $CO_2$, $H_2O$ and FMA, Los Gatos research, USA for $CH_4$) were 2.7, 24.3 and 5.4, respectively. At the same forest site, Kohonen et al. (2019) investigated fluxes of

carbonyl sulfide (COS) with a newer generation Aerodyne quantum cascade laser spectrometer (QCLS) (Aerodyne Research





Inc., Billerica, USA) that also measures mole fractions of $CO_2$ and $H_2O$. They reported high attenuation for their measurement setup (time constant of 0.68 s for their EC system), and high noise disturbance in their power spectra (their Fig. 7) with a SNR of about one. During the measurement period, COS showed small fluxes near the detection limit, causing uncertainty in the calculation of $H$. Thus, they calculated the time constant using the cospectra of $CO_2$, assuming that both fluxes are affected by

the same attenuation.

In addition, $N_2O$ fluxes measured over urban areas (Järvi et al., 2014), and $CO_2$ fluxes over lake ecosystems (Mammarella et al., 2015) can also be examples of low SNR conditions.

Given the wide range of SNR and attenuation conditions summarised above, we analysed only a limited range of SNR and attenuation. Also, the impact of a the system response time depends on the position of the spectral peak frequency which

changes not only with wind speed but also measurement height and surface roughness. Nevertheless, the analysis of the different approaches showed a systematic behaviour with respect to SNR level and attenuation conditions. This provides the opportunity to extend the results of this study beyond the examined values and guides the selection of the right method to find the relevant $H$.

Finally, the key constraint of the study was that we artificially simulate the various attenuation and SNR conditions, which

should be verified with *in situ* measurements.

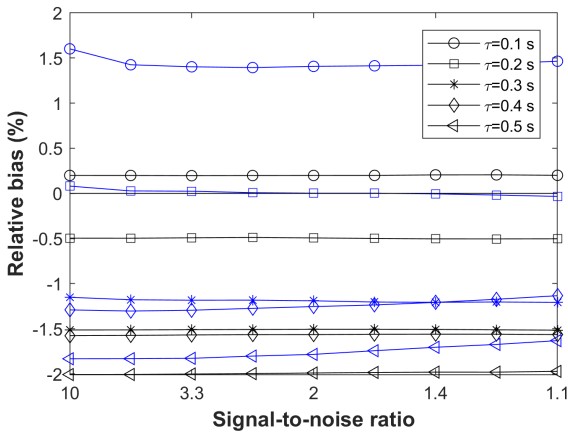

**Figure 8.** Relative biases of the cumulative fluxes derived with the various approaches compared with the reference flux as a function of SNR for different attenuation time scales (0.1-0.5 s), for $PSA_{I07}$ (blue) and $PSA_{A20}$ (black), calculated as, e.g., $100(F_{PSA_{I07}} - F_{REF})/F_{REF}$.

## 5    Conclusions

Here we investigated the limitations of two commonly used approaches to empirically estimated the eddy-covariance (EC) transfer function needed for the frequency response correction of measured fluxes by analysing a temperature flux time-series





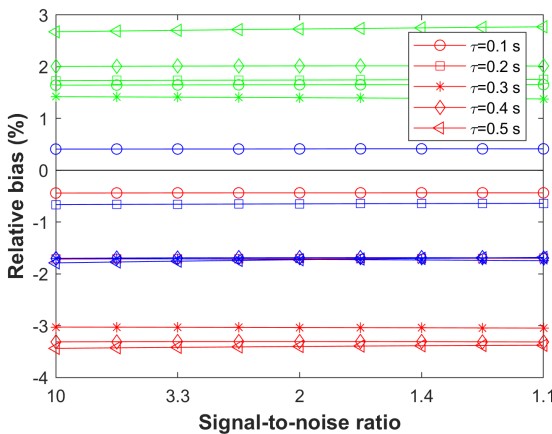

**Figure 9.** Relative biases of the cumulative fluxes derived with the various approaches compared with the reference flux as a function of SNR for different attenuation time scales (0.1-0.5 s), for $CSA_{\sqrt{H},sync}$ (blue), $CSA_{\sqrt{H}}$ (green) and $CSA_H$ (red), calculated as, e.g., $100(F_{CSA_{\sqrt{H},sync}} - F_{REF})/F_{REF}$.

which was synthetically degenerated mimicking slow-response setups and noisy sensors. The first approach (PSA) is based on the ratio of measured power spectra, while the second (CSA) is based on the ratio of measured cospectra. For PSA, we examined two alternative approaches of accounting for the white noise contribution to the power spectra: i.e. $PSA_{I07}$ and $PSA_{A20}$. The latter is newly introduced here and does not require noise removal prior to fitting of the response function. For CSA, we

examined three approaches: (1) utilising the transfer function ($H$) suggested by Horst (1997), i.e. $CSA_H$, (2) using $\sqrt{H}$, i.e. $CSA_{\sqrt{H}}$, and (3) implementing $\sqrt{H}$ with shifted $w$ time series via maximisation of the crosscovariance, i.e. $CSA_{\sqrt{H},sync}$. We generated an artificially dataset using $T$ with differing degrees of low-pass filtering (simulated damping) and additional random noise. The advantage of using artificial datasets is that the real values of frequency attenuation, SNR and physical time-lag are all known, allowing the precise comparison of the estimations and expected values. $PSA_{I07}$ overestimated the noise contri-

bution and consequently the signal loss and time constant for low attenuation conditions, but better performance was found as attenuation increases. The new $PSA_{A20}$ approach successfully estimated the time constants regardless of the attenuation and SNR level, identifying the noise and signal comprehensively, providing no bias and the lowest random uncertainty in the case of noisy data. However the approach assumed that the signal is contaminated by white noise, but this is not necessarily always the case. Hence, prior to the calculation of the time constant with this method, the nature of noise must be known. By

contrast, systematic differences were found between the results from three approaches based on cospectra, where the choice of the shape of $H$ differs in the literature. $CSA_H$ underestimated the time constants with increasing bias as attenuation increases, the accuracy of which was shown to be affected by neglecting the filtering effects on the quadrature spectrum, which cannot be avoided. $CSA_{\sqrt{H}}$ overestimated the time constants, the bias of which was mostly eliminated when time-lag correction was ap-





plied in $CSA_{\sqrt{H},sync}$. See the companion paper (Peltola et al., 2020) for a thorough discussion of the interdependence between time-lag estimation via cross-covariance maximisation and frequency response corrections.

We then examined the effect of the different approaches to estimate the time constants on the cumulative fluxes: fluxes corrected using the $PSA_{I07}$ based time constants showed the bias of $\pm 2\%$ in comparison with reference fluxes (see Sect.
3.3), where $PSA_{A20}$ showed similar behaviour, yet slightly better accuracy. By contrast, fluxes corrected using $CSA_H$ based time constants were underestimated by up to $4\%$, while $CSA_{\sqrt{H}}$ overestimated the fluxes by up to $\%3$, the bias of which was further reduced with $CSA_{\sqrt{H},sync}$, varying within $\pm 2\%$. The SNR did not affect the accuracy of either PSA or CSA approaches, alleviating concerns on EC flux measurements with low SNR level.

In summary, for the empirical estimation of parameters of $H$ of closed-path EC systems, our findings showed that $PSA_{A20}$
is the most accurate, precise and robust method when power spectra are used. This finding was independent of SNR and degree of attenuation. On the other hand, using the square root of $H$, which is a good approximation of the attenuation of cospectra in the presence of phase shifts (Peltola et al., 2020), provided the correct estimation of $\tau$ and $F_{corr}$ when cospectra are used, together with time-lag quantification by cross-covariance maximisation.

Finally, given the constraints of this study, we encourage additional studies based on the real attenuation and SNR conditions,
investigating also other type of noise contamination to provide a step forward in efforts to standardise the EC method, which is of great importance to avoid systematic biases of fluxes and improve comparability between different datasets.

*Code and data availability.*    Data to reproduce Figures 4, 5 6, 7, 8, 9, and code to demonstrate $PSA_{A20}$ are available in Aslan (2020).

## Appendix A:  Derivation of the $PSA_{A20}$

In this section, we derive the new approach for the PSA (i.e. $PSA_{A20}$) in detail. We assume that the measured $\chi$ contains two
independent and additive components: the attenuated turbulent signal and the noise. We further assume that scalar similarity holds (i.e. power spectra of all scalars follow a similar shape). After these assumptions the power spectrum ($S_\chi$) of $\chi$ can be written as:

$$S_\chi(f) = \frac{\sigma_\chi^2}{\sigma_T^2} H S_T(f) + S_{\chi,n}(f), \tag{A1}$$

where $\sigma_\chi^2$ and $\sigma_T^2$ are the variances of $\chi$ and $T$ related to turbulent signal (no attenuation or noise), $S_T(f)$ is the power spectrum
of $T$, $H$ describes the attenuation of $\chi$ due to imperfect instrumentation and $S_{\chi,n}$ is the noise in the $\chi$ measurements. Note that here the noise in the $T$ measurements was neglected as being small except in very low turbulence conditions (when eddy-covariance measurements become problematic anyway and fluxes are removed by the $u_*$ filter). Assuming that the instrument measuring $\chi$ can be approximated by a first-order linear sensor and reordering terms, this yields:

$$\frac{S_\chi(f)}{\sigma_\chi^2} = \frac{S_T(f)}{\sigma_T^2} F_n \frac{1}{1 + (2\pi f \tau)^2} + \frac{S_{\chi,n}(f)}{\sigma_\chi^2}, \tag{A2}$$





where the proportionality constant $F_n$ was introduced due to the effect that noise adds to $\sigma_\chi^2$ and thus biases the normalisation (see Ibrom et al., 2007a). The spectral density is best shown in log-log space as wide range of frequencies and spectral densities are well displayed (Stull, 2012), with natural frequency ($f$) on the x-axis, and spectral density multiplied with $f$ on the y-axis. After multiplication by $f$, Eq. (A2) becomes:

$$f\frac{S_\chi(f)}{\sigma_\chi^2} = f\frac{S_T(f)}{\sigma_T^2}F_n\frac{1}{1+(2\pi f\tau)^2} + f\frac{S_{\chi,n}(f)}{\sigma_\chi^2}, \tag{A3}$$

As described in Sect. 3.2.3, $f\frac{S_{\chi,n}(f)}{\sigma_\chi^2}$ can be approximated by a model that is linear in frequency in logarithmic space as $f\frac{S_{\chi,n}(f)}{\sigma_\chi^2} = alog(f) + log(b)$. In the case of white noise, where $a=1$, this results in a simplified linear model as $f\frac{S_{\chi,n}(f)}{\sigma_\chi^2} = log(f) + log(b)$, and in non-logarithmic space as $e^{log(f)+log(b)}$, which is $fb$. Substitution into Eq. (A3) yields:

$$f\frac{S_\chi(f)}{\sigma_\chi^2} = f\frac{S_T(f)}{\sigma_T^2}F_n\frac{1}{1+(2\pi f\tau)^2} + fb. \tag{A4}$$

It is worth mentioning that this method can be used to retrieve the variance of noise as well, since $b$ equals the ensemble averaged noise power spectra ($S_{\chi,n}$) divided with $\sigma_\chi^2$. However, we do not further examine it as it is not the main interest of this study.

**Appendix B: Assessment of the effect of quadrature spectrum on time constant estimation in the CSA**

In Sect. 4.2, we briefly presented the effect of $Q$ on time constant estimation in the $CSA_H$. In this section, the details are
provided.

     The transfer function used in this study (i.e. Eq. (3)) is the simplified version of the transfer function of Horst (2000) (their Eq. (5)) which neglects $Q$ and assumes that the sonic anemometer has a perfect frequency response. The theoretical transfer function including $Q$ can be written as:

$$H(f) = \frac{1+2\pi f\tau Q/Co}{1+(2\pi f\tau)^2}, \tag{B1}$$

where $Q$ is quadrature spectrum and $Co$ is the cospectrum of two variables and $\tau$ is the system time constant.

     For this particular analysis we used two $T$ datasets, the first of which is $D_1$ from Siikaneva and second of which is from Hyytiälä (see Peltola et al., 2020). We filtered the data by following the same procedure described in Sect. 3.2.1 with various $\tau_{LPF}$ ranging from 0.05 to 1 s. Then, we used different transfer functions (i.e. Eq. (3) and Eq. (B1)) to obtain time constants to examine the effect of $Q$.

Table B1 summarises the comparison of the time constants obtained with different transfer functions for $CSA_H$ for Siikaneva. Equation (3) underestimates $\tau_{CSA_H}$, while Eq. (B1) virtually eliminates this bias. A similar behaviour can be seen in Table B2 for the Hyytiälä dataset.

     The results of the two case studies combined indicate that $Q$ biases the time constant estimations, causing either under- or overestimation, depending on the sign of the term $2\pi f\tau Q/Co$ ratio in Eq. (B1). However, for gas measurements, an accurate





**Table B1.** Time constant estimations in order to assess the effect of quadrature spectrum in the cospectral approach ($CSA_H$) based on data from the Siikaneva site (dataset $D_1$). $\tau_{LPF}$ indicates the time constant of the first-order filter applied to the $T$ dataset; $\tau_{CSA_H}$ represents the time constants obtained via $CSA_H$ with two different transfer functions (i.e. Eq. (3) and Eq. (B1)), both of which are given with relative biases in the time constant values, respectively.

| $\tau_{LPF}$ (s) | $\tau_{CSA_H}$ with Eq. (3) (s) | Relative bias of Eq. (3) (%) | $\tau_{CSA_H}$ with Eq. (B1) (s) | Relative bias of Eq. (B1) (%) |
|---|---|---|---|---|
| 0.05 | 0.051 | 1.6 | 0.050 | -1.0 |
| 0.1 | 0.093 | -6.8 | 0.099 | -0.9 |
| 0.2 | 0.175 | -12.4 | 0.198 | -0.9 |
| 0.3 | 0.260 | -13.5 | 0.298 | -0.8 |
| 0.5 | 0.435 | -12.9 | 0.496 | -0.7 |
| 0.7 | 0.622 | -11.2 | 0.695 | -0.7 |
| 1 | 0.916 | -8.4 | 0.994 | -0.6 |

**Table B2.** Time constant estimations in order to assess the effect of quadrature spectrum in the cospectral approach ($CSA_H$) based on data data from the Hyytiälä site (see Peltola et al., 2020). $\tau_{LPF}$ indicates the time constant of the first-order filter applied to the $T$ dataset; $\tau_{CSA_H}$ represents the time constants obtained via $CSA_H$ with two different transfer functions (i.e. Eq. (3) and Eq. (B1)), both of which are given with relative biases in the time constant values, respectively.

| $\tau_{LPF}$ (s) | $\tau_{CSA_H}$ with Eq. (3) (s) | Relative bias of Eq. (3) (%) | $\tau_{CSA_H}$ with Eq. (B1) (s) | Relative bias of Eq. (B1) (%) |
|---|---|---|---|---|
| 0.05 | 0.056 | 12.6 | 0.050 | -0.6 |
| 0.1 | 0.104 | 4.3 | 0.099 | -0.7 |
| 0.2 | 0.190 | -5.1 | 0.199 | -0.7 |
| 0.3 | 0.275 | -8.4 | 0.298 | -0.7 |
| 0.5 | 0.449 | -10.3 | 0.496 | -0.8 |
| 0.7 | 0.622 | -11.2 | 0.695 | -0.8 |
| 1 | 0.874 | -12.9 | 0.992 | -0.8 |

calculation of the $Q/Co$ ratio is not feasible due to the physical time lag and high frequency attenuation induced by the sampling system.

Figures B1 and B2 show the ratio of normalized spectral densities of $Q$ and $Co$ of the unattenuated dataset of Siikaneva and Hyytiälä, respectively. Both ratios significantly deviate from zero across the frequency domain.





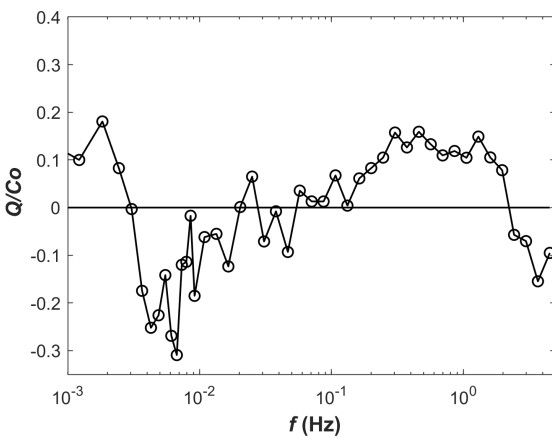

**Figure B1.** The ratio of normalised spectral densities of $Q$ and $Co$ of the unattenuated dataset of sonic temperature and vertical wind speed at Siikaneva, representing the ensemble average of dataset $D_1$ measured over the period from May to September 2013 during daytime with an average sensible heat flux of 114.3 W m$^{-2}$, friction velocity of 0.3 m s$^{-1}$, and wind speed of 2.1 m s$^{-1}$.

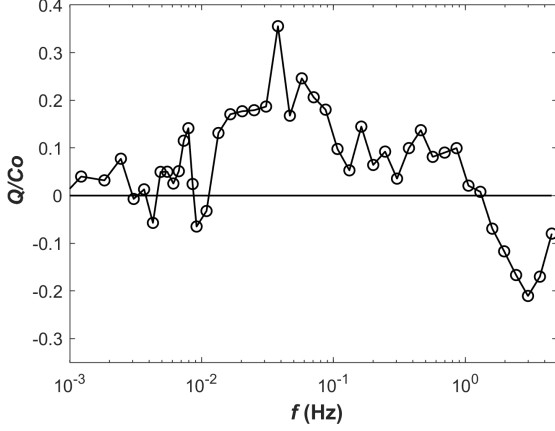

**Figure B2.** The ratio of normalised spectral densities of $Q$ and $Co$ of the unattenuated dataset of sonic temperature and vertical wind speed at Hyytiälä dataset measured over the period from May to August 2019 during daytime when the sensible heat flux was between 90 and 260 W m$^{-2}$, and the wind speed was between 2.1 and 3.7 m s$^{-1}$.

## Appendix C: Identification of instrumental noise

Here we perform a short analysis to characterise the type of noise in an example high frequency CH$_4$ time series by comparing the spectra of turbulent scalar with those of artificially generated white and blue noise.





Methane fluxes from an upland forest site can be considered to be greatly affected by instrumental noise because fluxes are small and near the detection limit. Therefore, we used a forest methane mixing ratio dataset to identify the type of noise by comparing its spectra with the spectra of white and blue noise which was artificially generated and with the same standard deviation of the methane dataset.

The data were collected at the SMEAR II station (Station for Measuring Forest Ecosystem- Atmosphere Relationships), Hyytiälä, Southern Finland ($61^o51'$ N, $24^o17'$ E; 181m a.s.l.) which is Class 1 ICOS ecosystem station. The station is surrounded by extended areas of coniferous forests and the EC tower is located in a 55-year-old (in 2017) Scots pine ($Pinus sylvestris$ L.) forest with a dominant tree height of 19 m. The measurements were performed with 10 Hz sampling frequency at a height of 23 m, i.e. approximately 4 m above the forest canopy. A fast response laser absorption spectrometer (G2311-f, Picarro,
USA) was used to measure $CH_4$ mixing ratio.

Power spectra of methane were calculated using measurements from the 10 July at 12:00-14:30. The high frequency range of the normalised frequency weighted $CH_4$ power spectrum follows the $f^{+1}$ power law scaling, which is consistent with white noise and inconsistent with, e.g., blue noise (Fig. C1).

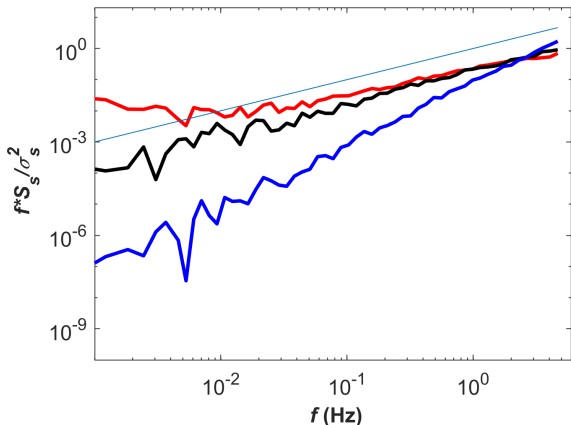

**Figure C1.** Normalized spectra of methane concentration (red), white noise (black), and blue noise (blue) ($fS_s/\sigma_s^2$), all of which have the same standard deviation. Solid straight line is $f^{+1}$ line.

## Appendix D: The site specific cospectral model used in flux calculations

The cospectral model used in flux calculations for small fluxes (i.e. absolute value of sensible heat flux smaller than 15 W m$^{-2}$) was calculated following Horst (1997)):

$$\frac{fCo(f)}{\overline{w'\chi'}} = \frac{2}{\pi}\frac{n/nm}{1+(n/nm)^2},$$ (D1)


where $n$ is the normalised frequency, $nm$ is the cospectral peak frequency, which is derived from in situ measurements in Siikaneva as:

$$nm = \begin{cases} 0.09, & \frac{z-d}{L} \leq 0, \\ 0.09(1+4.5(\frac{z-d}{L})^{0.78}), & \frac{z-d}{L} > 0, \end{cases} \tag{D2}$$

where $z$ is the measurement height, $d$ is the displacement height, and $L$ is the Obkuhov length, a measure of atmospheric stability.

*Author contributions.* IM and OP designed the study and TA processed and analyzed the data. UR investigated the distortion caused by quadrature spectrum in CSA. UR, TA and AI investigated the limitations of noise removal procedure of $PSA_{I07}$. AI developed the $PSA_{A20}$ following an idea that emerged during a stimulating discussion between all authors, and investigated the square-root conundrum in CSA. TA wrote the manuscript with contributions from all co-authors.

*Competing interests.* The authors declare that they have no conflict of interest.

*Acknowledgements.* This study was supported by ICOS and through the H2020 RINGO project of the European Commission (grant 730944). TA is grateful to the Finnish National Agency For Education and The Vilho, Yrjö and Kalle Väisälä foundation for their kind support for funding. OP is supported by the postdoctoral researcher project (decision 315424) funded by the Academy of Finland, and EN acknowledges support by the Natural Environment Research Council award number NE/R016429/1 as part of the UK-SCAPE programme delivering National Capability.





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
