# Peer review of "The high frequency response correction of eddy covariance fluxes. Part 1: an experimental approach for analysing noisy measurements of small fluxes"

_Atmospheric Measurement Techniques, 2020_

## Referee Comment (RC1) · Marc Aubinet (Referee) · 15 Jan 2021

marc.aubinet@ulg.ac.be Received and published: 15 January 2021

**General comments**

This paper is the first of two papers discussing different spectral corrections procedures for low pass filtering effects in eddy covariance systems. I read and reviewed both and found the present paper difficult to understand without reading the second (Peltola et al, also on AMT discussions). I thus recommend the authors to change the paper order and put the Peltola paper, which better stands alone, in first position and this one in

second. I also made my reviews in this order and, for people who would be interested, I also recommend to read my review of the Peltola paper before this one.

This paper addresses two questions related to low pass frequency corrections for eddy covariance systems: one is the problem of power spectra contamination by white noise and the resulting difficulty to determine transfer function time constants when using the spectral approach (PSA); the second is the impact of transfer function shape and of time lag when using a cospectral approach (CSA).

Having never applied the PSA on low signal to noise ratios personnaly, I have no direct experience of the impact of noise on power spectra but I can imagine that this problem may be critical under low SNR. The method proposed by the authors seems to provide more accurate and less dispersed time constant estimates than the classical method proposed by lbrom et al. 2007 (I suppose that Andreas supports this, as he is co-author of the paper). However, I didn't find the method very convincing as flux estimates obtained with this new approach did not appear much more accurate (Figure 8). In addition, I wonder about the feasibility of applying routinely this approach on "real world data" as the fit has to provide two parameters which could create convergence problems.

Above all, this makes me again wonder why one persists to follow the PSA while the CSA approach is not affected by noise.

I'm much more reluctant about the second comparison. The authors apply the CSA and compare three approaches, two based on non-synchronized cospectra and using either a Lorentzian curve or its square root and the third based on synchronized (time lag adjusted) cospectra using the square root of a Lorentzian. I was first puzzled by the use of non-synchronised cospectra, that appears a priori nonsense. In the present case, anyway, as the original (not attenuated) time series are not lagged, I suppose that the first approach (CSAH) could be thought as an application of the classical Lorentzian transfer function on a set on which the time lag introduced by signal attenuation would

have been ignored. The third one (CSA sqrt(H), sync) would then be those taking both attenuation and its derived time lag into account, following an approach described in the Peltola paper (Method 2). This comparison would then show that ignoring the time lag due to low pass filtering would lead to a time constant underestimation. If my interpretation is correct, I think that it should at least be explained by the authours.

Besides this, I don't see the interest of the second approach (CSA sqrt(H)) as it does not correspond to any used methodology. Contrary to what the authors suggest (P5L21), this comparison does not address the debate on transfer function shape: indeed the real question, very well synthesized by Peltola, was to determine, in the PSA, which transfer function should be applied on cospectra: the function itself or its square root (with the same time constant). This is not what was tested here as the Lorentzian and its square root were separately adjusted on the same data set resulting in a quite trivial result, i.e. the time constant of the CSA sqrt(H) is about twice those of the CSA H. An approach fitting a Lorentzian on synchronised time (CSA H, sync) would probably be more relevant as it would mimic the Method 1 of Peltola presently (and erroneously) recommended by the ICOS protocols. Anyway, I think that these comparisons are of limited interest as they overlap with results of Peltola.

Finally, I found both analyses (as well on PSA as on CSA) too much focused on time constants, which are not an objective per se when applying spectral correction procedures, and not enough on correction factors (not presented in the study) or fluxes. Correction factors are not presented and only cumulated fluxes are presented and (too) shortly discussed. I think that the real efficiency of the approaches can better be evaluated by looking half hourly fluxes and I suggest the authors to look at the regressions between half hourly fluxes obtained with the different approaches.

In the whole, the paper presents some promising results but some of the proposed comparisons are not relevant to my opinion and some analyses are insufficiently developed and not enough focused on the real poroduct of the spectral correction, i.e., the correction factors and the fluxes.

СЗ

Besides this, the paper is generally well written and presented but there are still some presentation problems that I point in the specific comments below. In conclusion, I think that the paper needs a major revision before publication.

Specific comments

P2L11 and P4L13: In the frequency space these operations are not convolutions but multiplications (a convolution in the time space corresponds to a multiplication in the frequency space and conversely).

P3L10: suppress the "be"

P3L11: Is this really critical? As the time lag is mainly determined by the set up, it could be determined on periods of larger flux and extrapolated. On the other hand, it's true that for small SNR, a time lag estimated by covariance maximisation, would systematically select the time lag associated to the highest flux and would not necessarily correspond to a "physical" maximum, which could lead to bias small fluxes.

P3L13: Yes but the paper showed that the impact on correction factors and fluxes was not critical if adequately accounted.

P5L21: See my general comments above. I think that the proposed experiment does not bring any relevant argument to this debate.

P5L29 and foll.: You also point below (P9L21) that, in the lbrom procedure, the boundaries for regression fitting are fixed by eye. This should be specified here as you consider this as a limit of the method.

P6L9: What does mean a "y axis intercept" in a log scale? In your case, this axis does even not appear in the figure!

P7L21-24: I had difficulties to understand this paragraph but the problem is maybe simply that you should refer to Eq (7) rather than Eq (4) on L24.

P7L32-P8L2: See general comments above.

P9L17-23: This is somewhat a repetition of denoising description above (P5L29 and foll.) but only the I07 method is described. What about the A20 method? Clarify and avoid overlapping.

P12 Fig 4, P15 Fig 6: Spectra and cospectra were presented in function of the real frequency. Did you take the possible cospectral shift with wind velocity when taking ensemble averages?

P13Fig5: Red line and shaded area are confounded.

P14L10: the term "raw" is maybe not very well adapted here as it represents in fact the ideal cospectrum, without attenuation and noise.

P14L10: Was the cospectrum based on synchronized time series or not?

P14L10: It is in fact not so evident from the figure that noise contamination does not affect the cospectra shape. Do you refer to the fact that no linear increase is observed at high frequency? This could be specified.

P14L12 and foll : See my general comment.

P14L27: Yes but this is a problem that affects time constant determination but not correction factors and fluxes, as shown by Peltola.

P15L2: As I also pointed in my review of the Peltola paper, I'm not convinced by the use of relative errors on cumulated fluxes to evaluate the performance of correction methods. On one hand, relative errors are often not informative (the relative error on a zero flux would be infinite !), on the other hand computing errors on cumulated fluxes only would hinder error compensation (beteen night and day, for example). I prefer a comparison between half hourly fluxes (by taking the regression slope, for example).

P15L5: In view of Figure 8, it is not clear to me that PSA20 is more accurate than PSI07. I just see that it underestimates systematically the flux value but there is no clue that the bias is smaller.

P15L6: Very close (<1%) is only the case for the low time constants (0.1 and 0.2 s). For higher time constants, it anyway reaches 2%.

---

## Referee Comment (RC2) · George Burba (Referee) · 15 Jan 2021

This manuscript could result in a significant improvement in the calculations of eddy covariance fluxes of $H_2O$ and $CO_2$ over the ecosystems and during periods when the fluxes are small (e.g., deserts and areas sublimation for $H_2O$, winter and offseason for $CO_2$ etc.), and in the calculations of eddy covariance fluxes of $CH_4$, $N_2O$, Ammonia, Isotopes and other "small-flux" species in most situations and cases.

Particular improvements should be observed when constructing long-term budgets,

when long periods of extremely low uptakes or releases are typically followed by short periods of large releases.

Another advantage of the proposed technique is the ability to reasonably automate or semi-automate it so that numerous non-micrometeorological researchers who measure small fluxes increasingly frequently can take advantage of this new improvement.

I have included over 200 suggestions, as track changes and comments, in the attached file. Most are minor, however few are major:

1. It would be very helpful if authors could illustrate the actual fit and how it is different vs and convention technique (see specific comments). A simplified graphical example or two may go a long way. Reader needs some feel for what is changing and how much.

2. Examples of correction factors from the simulated dataset, and also from a few real-life datasets would also be helpful.

3. The need for non-time-lag adjusted cospectra should be explained very carefully. I suggested some ideas in the attached. Without such explanations, the two non-time-lag adjusted cospectral approached seem like artificial issues created solely for the purpose of solving them.

4. In the Conclusions section, it may be very useful to provide an assessment on the ease and reliability of the automaton for each of the compared techniques. I have included some ideas. The full impact of the newly proposed technique would only happen if a broad community accepts it and start using it. Automation or semi-automation is one of the keys to such acceptance and use.

I am not sure if these suggestions require Minor or Major Revision. Probably a medium one :) I have indicated the Minor Revision but would let authors and Editor decide on this.

Please also note the supplement to this comment:
https://amt.copernicus.org/preprints/amt-2020-478/amt-2020-478-RC2-
supplement.zip

---

## Author Comment (AC2) · 7 Apr 2021

Manuscript: Title: The high frequency response correction of eddy covariance fluxes. Part 1: an experimental approach for analysing noisy measurements of small fluxes

Author(s): Toprak Aslan et al.

MS No.: amt-2020-478

MS type: Research article

Reviewer's comments are shown in **bold**, while authors' responses are in red.

Reviewer 2 # George Burba

We thank the referee for his positive comments and appreciate the changes, correcting typos and improving English, and making the flow simpler and clearer for the general audience. Since there are two different files containing comments, in order to prevent repeating the same answers and provide clarity, we combined the comments from both pdf and word file, and summarized them below under the four different bullet points already highlighted by the referee. The comments from the word file are shown with page and line number in the manuscript. All editing corrections are accepted, hence they are not shown here.

1. It would be very helpful if authors could illustrate the actual fit and how it is different vs and convention technique (see specific comments). A simplified graphical example or two may go a long way. Reader needs some feel for what is changing and how much.

- P6L12 (from the word file): So, in the beginning of this section, authors explain the existing procedure and demonstrate it graphically in Figure 1.

Then authors discuss the deficiency of the resulted black line due to the separate fitting of the blue line, and propose a new combined way to fit.

Here, it may be very illustrative and helpful for the reader's understanding to present a similar plot (1B) with a graphical explanatory illustration of the new proposed procedure and show the difference of a new black line from the old black line.

It may be hard to see the difference over the entire frequency scale, so perhaps it could be shown just for the portion of f(Hz) from 10(-2) to 5 Hz?

- P13L4 (from the word file): If it would be possible to show this in Figure 1B in a graphical explanatory way, it would really made it much easier to understand the new procedure.
- Thank you for the suggestion. We updated Section 2.2, and Figure 1 (shown below with its caption). We preferred not putting another figure, but enhancing the current one. All steps in PSA107, and PSAA20 with advantages and disadvantages are explained in detail via referring to updated Figure 1.

Figure 1. A diagram illustrating fitting procedures for PSA methods. Shown are the spectra of unattenuated and noise-free temperature (red line), and spectra of low-pass filtered and noisy scalar (blue-solid line) and after (black line) noise removal. For PSA107, the noise is detected via fitting a line (blue-dashed) to the high-frequency end of noisy scalar over the frequency range highlighted. Then, it is extended towards lower

frequencies, and subtracted from the noisy spectrum, yielding noise-free spectra. Later, the time constant is calculated via fitting Eq. (4) to noise-free spectra over the frequency range highlighted. For PSAA20, the time constant is obtained from one comprehensive fitting Eq. (7) to noisy spectra over the whole frequency range highlighted.

**2.** Examples of correction factors from the simulated dataset, and also from a few real-life datasets would also be helpful.

- P13L5 (from the word file): Why? For simulated attenuation and SNR maybe, but for real-life data
  I am not sure.
  - We added a real-world data, i.e. CO2 from Siikaneva site, to demonstrate the performance of PSAA20 in comparison with PSAI07 and CSAsqrt H, sync. The results are shown and discussed in Section. 4.4.

What are correction factors for this dataset?

 Following also the parallel request from the referee 1, Marc Aubinet, we added a figure (Fig. 2 shown below) illustrating the correction factors for a single case (i.e., τ=0.3 s, SNR=2).

---

## Author Response (AR1)

Point-by-point responses to the reviews including all relevant changes made in the manuscript are summarized below. Reviewer's comments are shown in **bold black**, while authors' responses are in red. The changes mentioned below are visible in the revised manuscript with track-changes.

**Reviewer 1 # Marc Aubinet:**

**General comments**

**This paper is the first of two papers discussing different spectral corrections procedures for low pass filtering effects in eddy covariance systems. I read and reviewed both and found the present paper difficult to understand without reading the second (Peltola et al, also on AMT discussions). I thus recommend the authors to change the paper order and put the Peltola paper, which better stands alone, in first position and this one in second. I also made my reviews in this order and, for people who would be interested, I also recommend to read my review of the Peltola paper before this one.**

**This paper addresses two questions related to low pass frequency corrections for eddy covariance systems: one is the problem of power spectra contamination by white noise and the resulting difficulty to determine transfer function time constants when using the spectral approach (PSA); the second is the impact of transfer function shape and of time lag when using a cospectral approach (CSA).**

**Having never applied the PSA on low signal to noise ratios personally, I have no direct experience of the impact of noise on power spectra but I can imagine that this problem may be critical under low SNR. The method proposed by the authors seems to provide more accurate and less dispersed time constant estimates than the classical method proposed by Ibrom et al. 2007 (I suppose that Andreas supports this, as he is co-author of the paper). However, I didn't find the method very convincing as flux estimates obtained with this new approach did not appear much more accurate (Figure 8).**

**In addition, I wonder about the feasibility of applying routinely this approach on "real world data" as the fit has to provide two parameters which could create convergence problems.**

**Above all, this makes me again wonder why one persists to follow the PSA while the CSA approach is not affected by noise.**
**I'm much more reluctant about the second comparison. The authors apply the CSA and compare three approaches, two based on non-synchronized cospectra and using either a Lorentzian curve or its square root and the third based on synchronized (time lag adjusted) cospectra using the square root of a Lorentzian. I was first puzzled by the use of non-synchronised cospectra, that appears a priori nonsense. In the present case, anyway, as the original (not attenuated) time series are not lagged, I suppose that the first approach (CSAH) could be thought as an application of the classical Lorentzian transfer function on a set on which the time lag introduced by signal attenuation would have been ignored. The third one (CSA sqrt(H), sync) would then be those taking both attenuation and its derived time lag into account, following an approach described in the Peltola paper (Method 2). This comparison would then show that ignoring the time lag due to low pass filtering would lead to a time constant underestimation. If my interpretation is correct, I think that it should at least be explained by the authours.**

**Besides this, I don't see the interest of the second approach (CSA sqrt(H)) as it does not correspond to any used methodology. Contrary to what the authors suggest (P5L21), this comparison does not address the debate on transfer function shape: indeed the real question, very well synthesized by Peltola, was to determine, in the PSA, which transfer function should be applied on cospectra: the function itself or its**

square root (with the same time constant). This is not what was tested here as the Lorentzian and its square root were separately adjusted on the same data set resulting in a quite trivial result, i.e. the time constant of the CSAsqrt(H) is about twice those of the CSAH. An approach fitting a Lorentzian on synchronised time (CSA H, sync) would probably be more relevant as it would mimic the Method 1 of Peltola presently (and erroneously) recommended by the ICOS protocols. Anyway, I think that these comparisons are of limited interest as they overlap with results of Peltola.

Finally, I found both analyses (as well on PSA as on CSA) too much focused on time constants, which are not an objective per se when applying spectral correction procedures, and not enough on correction factors (not presented in the study) or fluxes. Correction factors are not presented and only cumulated fluxes are presented and (too) shortly discussed. I think that the real efficiency of the approaches can better be evaluated by looking half hourly fluxes and I suggest the authors to look at the regressions between half hourly fluxes obtained with the different approaches.

In the whole, the paper presents some promising results but some of the proposed comparisons are not relevant to my opinion and some analyses are insufficiently developed and not enough focused on the real product of the spectral correction, i.e., the correction factors and the fluxes.

Besides this, the paper is generally well written and presented but there are still some presentation problems that I point in the specific comments below. In conclusion, I think that the paper needs a major revision before publication.

- General Response: We thank the referee for the comments. The hypothesis of the present study was "the success of the PSA and CSA usage in frequency response corrections depends on the attenuation condition and the level of signal-to-noise ratio", which was successfully investigated and concluded. Indeed, the study mainly focused on time constant, and cannot provide holistic understanding on spectral correction as highlighted by the referee. This is partly due to the lack of a robust method for calculating correction factors as shown by the companion paper (Peltola et al. 2021). As a result, the effect of variations in time constant estimations cannot be clearly seen in correction factor calculations. In other words, the methods used for time constant estimation does not have a significant effect on corrections factors, hence final fluxes. In the analysis of the originally submitted manuscript, the correction factors used to correct the artificially attenuated fluxes were calculated following the Fratini et al. (2012) approach and compared with the unattenuated fluxes. When using such an approach, the reference fluxes were not fully comparable with the corrected fluxes due to the low-pass filtering related phase shift effects, as shown in the companion paper (Peltola et al., 2021). That prevents showing the sole effect of the deviation in time constant estimation on correction factors and fluxes. Thus, we changed the calculation of reference fluxes, so that it would be calculated with Eq. 11 similar to other methods. For this, the time constants used for low-pass filtering is implemented in $H_{emp}$. It is worth noting that with this change the variation in correction factors would solely reflect the variation in time constant. That enables a good statistical comparison, however does not reflect the shortcomings of the method, Fratini et al. (2012) when used in real life data. The relevant results are shown below (Figs. 1, 2 and 3). These findings are consistent with the observed biases on time constant estimation, meaning that where the time constant and the low-pass filtering were overestimated (e.g. with the $PSA_{I07}$ especially for tau=0.1 s), the spectral correction factor and thus the fluxes were overestimated, too. The correction factors of a case with tau=0.3 s and SNR=2 is also shown below (Fig. 3). We added the figure to the manuscript.

[Figure]

Fig. 1 Relative biases of the cumulative fluxes derived with the various approaches compared with the reference flux as a function of SNR for different attenuation time scales (0.1-0.5 s), for $PSA_{I07}$ (blue) and $PSA_{A20}$ (black), calculated as, e.g., $100(F_{PSAI07} - F_{REF})/ F_{REF}$.

[Figure]

Fig. 2 Relative biases of the cumulative fluxes derived with the $CSA_{sqrt(H),sync}$ compared with the reference flux as a function of SNR for different attenuation time scales (0.1-0.5 s), calculated as, e.g., $100(F_{CSAsqrt(H),sync} - F_{REF})/ F_{REF}$.

[Figure]

Fig 3. Correction factors ($F_{corr}$) of half-hourly fluxes calculated with different approaches, i.e. $PSA_{A21}$ (black cross), $PSA_{I07}$ (blue), $CSA_{sqrt(H),sync}$ (green) and the reference (red) for the case with tau=0.3 s and SNR of two.

- Regarding using the CSA variations without time-lag corrections, following the referee's suggestion we removed the $CSA_H$ and $CSA_{sqrt(H)}$ as it causes confusion and does not provide any contribution to the discussion in the literature. $CSA_{sqrt(H), sync}$ is the only CSA method used in the study. Additionally, we prefer not implementing $CSA_{H, sync}$ as suggested by the referee to prevent overlapping the findings in the companion paper. Moreover, we removed the Appendix B. "Assessment of the effect of quadrature spectrum on time constant estimation in the CSA" and the relevant info in the Section 4.2. "Time constant estimation with CSA'' from the manuscript as it overlaps with the theoretical explanation of the phase shift effect in the companion paper.

- After these changes, in order to provide better flow, we changed the order of the companion papers as suggested by the referee.

- A concern related to the applicability of the new PSA ($PSA_{A20}$) method in real-world data is raised by the both referees. Thus, in order to demonstrate the performance of the $PSA_{A20}$ and compare it with $PSA_{I07}$ and $CSA_{sqrt(H), sync}$ , we processed a real-world data, i.e. $CO_2$ fluxes from Siikaneva peatland site, for time constant calculation and added the results in the revised manuscript.

**Specific comments**

**P2L11 and P4L13: In the frequency space these operations are not convolutions but multiplications (a convolution in the time space corresponds to a multiplication in the frequency space and conversely).**

- Response 1: The term "convolution" is removed from the manuscript at P2L11, and replaced with the term "multiplication" at P4L13.

**P3L10: suppress the "be"**

- Response 2: Done.

**P3L11: Is this really critical? As the time lag is mainly determined by the set up, it could be determined on periods of larger flux and extrapolated. On the other hand, it's true that for small SNR, a time lag estimated by covariance maximisation, would systematically select the time lag associated to the highest flux and would not necessarily correspond to a "physical" maximum, which could lead to bias small fluxes.**

- Response 3: We agree with the referee, but argue that this issue is worth mentioning in this context due to its various implications on flux estimates, as discussed in Langford et al. (2015). Estimation of time lags during periods when the fluxes are large and then extrapolating to low flux periods works (see for example Rannik et al. (2015)) but naturally only if high fluxes are observed during the measurement period.

**P3L13: Yes but the paper showed that the impact on correction factors and fluxes was not critical if adequately accounted.**

- Response 4: This is explained in the general response above.

**P5L21: See my general comments above. I think that the proposed experiment does not bring any relevant argument to this debate.**

- Response 5: This is related to the implementing $CSA_H$ and $CSA_{sqrt(H)}$, which are removed from the manuscript. Please see the general response above.

**P5L29 and foll.: You also point below (P9L21) that, in the Ibrom procedure, the boundaries for regression fitting are fixed by eye. This should be specified here as you consider this as a limit of the method.**

- Response 6: We thank the referee for the comment. We specified the shortcomings of the $PSA_{I07}$ in Section 2.2.

**P6L9: What does mean a "y axis intercept" in a log scale? In your case, this axis does even not appear in the figure!**

- Response 7: Due to this comment we tried to clarify the text and rewrote this part of the manuscript as "where b is the ratio between noise variance and variance used to normalize the \chi power spectrum" (see Appendix A).

**P7L21-24: I had difficulties to understand this paragraph but the problem is maybe simply that you should refer to Eq (7) rather than Eq (4) on L24.**

- Response 8: Thank you for the correction. As you explained, there is a mistake with equation numbers. We now refer to Eq (7).

**P7L32-P8L2: See general comments above.**

- Response 9: This is again related to the implementing $CSA_H$ and $CSA_{sqrt(H)}$, which are removed from the manuscript. Please see the general response above.

**P9L17-23: This is somewhat a repetition of denoising description above (P5L29 and foll.) but only the I07 method is described. What about the A20 method? Clarify and avoid overlapping.**

- Response 10: The referee is right about the repetition of noise removal procedure. Thus, we removed the Section 3.2.3 "Noise removal". We moved the important information about the frequency range for noise removal procedure to Section 4.1 "Time constant estimation with PSA".

- More details about the procedures of $PSA_{A20}$ are now given in Section 2.2., and the schematic figure (Fig 1.) is enhanced accordingly.

**P12 Fig 4, P15 Fig 6: Spectra and cospectra were presented in function of the real frequency. Did you take the possible cospectral shift with wind velocity when taking ensemble averages?**

- Response 11: No we did not. It is hard to determine the peak of the individual cospectras due to their shapes. Furthermore, the aim of these figures is to show the difference between attenuated

and not attenuated (co-)spectra and wind speed does not alter that difference (as the transfer function calculated with Eq. 3 in the manuscript does not depend on wind speed).

**P13Fig5: Red line and shaded area are confounded.**

- Response 12: Indeed, the red line is not clearly seen. Thus, we changed the color of the line representing the median from red to black.

**P14L10: the term "raw" is maybe not very well adapted here as it represents in fact the ideal cospectrum, without attenuation and noise.**

- Response 13: We made the necessary changes via replacing "raw" with "unattenuated and noise free".

**P14L10: Was the cospectrum based on synchronized time series or not?**

- Response 14: Yes, it was synchronized. We made it clear via modifying the sentence as follows, "...illustrates **the time-lag corrected** cospectra of three low-pass filtered cases…".

**P14L10: It is in fact not so evident from the figure that noise contamination does not affect the cospectra shape. Do you refer to the fact that no linear increase is observed at high frequency? This could be specified.**

- Response 15: Here we refer to no significant deviations that would affect the time constant estimations. We replaced the sentence "It shows that noise contamination does not affect the shape of the cospectra" with "It shows that the white noise contamination did not cause linear increase in the high-frequency end of the cospectra, enabling the time-constant calculation without additional procedure related to noise removal".

**P14L12 and foll : See my general comment.**

- Response 16: The CSA approaches without time-lag correction referred here are removed from the manuscript as stated in Response 5 and general response above.

**P14L27: Yes but this is a problem that affects time constant determination but not correction factors and fluxes, as shown by Peltola.**

- Response 17: Please see the general response above.

**P15L2: As I also pointed in my review of the Peltola paper, I'm not convinced by the use of relative errors on cumulated fluxes to evaluate the performance of correction methods. On one hand, relative errors are often not informative (the relative error on a zero flux would be infinite !), on the other hand computing errors on cumulated fluxes only would hinder error compensation (between night and day, for example). I prefer a comparison between half hourly fluxes (by taking the regression slope, for example).**

- Response 18: We tried showing the biases by comparing the half-hourly fluxes corrected with different approaches with the reference fluxes. An example of such comparison for a specific case (with tau=0.3s and SNR=2) is shown below (Fig 4). Since the biases are very small, the differences are not visible when showing them in a scatter diagram, hence the statistics of the regression analysis are not informative as well. In the Fig. 4 below, the reference line and the regression line are confounded due to similarity and the regressions slopes are 0.9856, 1.0001 and 0.9979, respectively.  Indeed, the relative bias is not the best way for the comparison, but we believe that it is the only representative method for our findings, thus we would like to use it in the manuscript.

Note also that as the spectral corrections are multiplicative (not additive) they do not e.g. change the sign of the flux and hence showing their performance in a relative sense is natural.

[Figure]

Fig. 4 Comparison of half-hourly fluxes calculated with different approaches (i.e.PSA$_{I07,}$ PSA$_{A20}$ and CSA$_{sqrt(H),}$ $_{sync}$ , respectively)  with the reference fluxes. The reference 1:1 line is shown with a dashed-black line, while the regression line is shown with solid-red line.

**P15L5: In view of Figure 8, it is not clear to me that PSA20 is more accurate than PSI07. I just see that it underestimates systematically the flux value but there is no clue that the bias is smaller.**

- Response 19: We now updated the Figure 8 and 9 (see general response above).

**P15L6: Very close (<1%) is only the case for the low time constants (0.1 and 0.2 s). For higher time constants, it anyway reaches 2%.**

- Response 20: We changed the calculation of the correction factors for the reference fluxes, hence the relevant figures (8 and 9) are updated. Please see the general response above.

References

Langford, B., Acton, W., Ammann, C., Valach, A., & Nemitz, E. (2015). Eddy-covariance data with low signal-to-noise ratio: time-lag determination, uncertainties and limit of detection. Atmospheric Measurement Techniques, 8(10), 4197-4213.

Rannik, Ü., Haapanala, S., Shurpali, N. J., Mammarella, I., Lind, S., Hyvönen, N., ... & Vesala, T. (2015). Intercomparison of fast response commercial gas analysers for nitrous oxide flux measurements under field conditions. Biogeosciences, 12(2), 415-432.

**Reviewer 2 # George Burba:**

We thank the referee for his positive comments and appreciate the changes, correcting typos and improving English, and making the flow simpler and clearer for the general audience. Since there are two different files containing comments, in order to prevent repeating the same answers and provide clarity, we combined the comments from both pdf and word file, and summarized them below under the four different bullet points already highlighted by the referee. The comments from the word file are shown with page and line number in the manuscript. All editing corrections are accepted, hence they are not shown here.

**1. It would be very helpful if authors could illustrate the actual fit and how it is different vs and convention technique (see specific comments). A simplified graphical example or two may go a long way. Reader needs some feel for what is changing and how much.**

- **P6L12 (from the word file): So, in the beginning of this section, authors explain the existing procedure and demonstrate it graphically in Figure 1.**

    **Then authors discuss the deficiency of the resulted black line due to the separate fitting of the blue line, and propose a new combined way to fit.**

    **Here, it may be very illustrative and helpful for the reader's understanding to present a similar plot (1B) with a graphical explanatory illustration of the new proposed procedure and show the difference of a new black line from the old black line.**

    **It may be hard to see the difference over the entire frequency scale, so perhaps it could be shown just for the portion of *f(Hz)* from 10^(-2) to 5 Hz?**

- **P13L4 (from the word file): If it would be possible to show this in Figure 1B in a graphical explanatory way, it would really made it much easier to understand the new procedure.**

- Thank you for the suggestion. We updated Section 2.2, and Figure 1 (shown below with its caption). We preferred not putting another figure, but enhancing the current one. All steps in $PSA_{I07}$, and $PSA_{A20}$ with advantages and disadvantages are explained in detail via referring to updated Figure 1.

[Figure]

Figure 1. A diagram illustrating fitting procedures for PSA methods. Shown are the spectra of unattenuated and noise-free temperature (red line), and spectra of low-pass filtered and noisy scalar (blue-solid line) and after (black line) noise removal. For PSA$_{I07}$, the noise is detected via fitting a line (blue-dashed) to the high-frequency end of noisy scalar over the frequency range highlighted. Then, it is extended towards lower frequencies, and subtracted from the noisy spectrum, yielding noise-free spectra. Later, the time constant is calculated via fitting Eq. (4) to noise-free spectra over the frequency range highlighted. For PSA$_{A20}$, the time constant is obtained from one comprehensive fitting Eq. (7) to noisy spectra over the whole frequency range highlighted.

**2. Examples of correction factors from the simulated dataset, and also from a few real-life datasets would also be helpful.**

- **P13L5 (from the word file): Why? For simulated attenuation and SNR maybe, but for real-life data I am not sure.**
    - We added a real-world data, i.e. $CO_2$ from Siikaneva site, to demonstrate the performance of PSA$_{A20}$ in comparison with PSA$_{I07}$ and CSA$_{sqrt\ H,\ sync}$. The results are shown and discussed in Section. 4.4.
- **What are correction factors for this dataset?**
    - Following also the parallel request from the referee 1, Marc Aubinet, we added a figure (Fig. 2 shown below) illustrating the correction factors for a single case (i.e., τ=0.3 s, SNR=2).

[Figure]

Figure 2. Correction factors ($F_{corr}$) of half-hourly fluxes calculated with different approaches, i.e. PSA$_{A21}$ (black cross), PSA$_{I07}$ (blue point), CSA$_{sqrt(H),sync}$ (green point) and the reference (red point) for the case with tau=0.3 s and SNR of two.

> Is it possible to add examples of correction factors for other independent datasets for both PSAI07 and PSAA20?

> o The estimated time constants for a real-world data are added to the manuscript. However, we didn't further process the fluxes, hence $F_{corr}$ is not provided as the resulting figure would look similar to Figure 2. shown above.

- **P14L10 (from the word file): Only for truly "white" simulated noise.**
  - o We replaced the relevant sentence "It shows that noise contamination does not affect the shape of the cospectra" with "It shows that the white noise contamination did not cause linear increase in the high-frequency end of the cospectra, enabling the time-constant calculation without additional procedure related to noise removal".

**3. The need for non-time-lag adjusted cospectra should be explained very carefully. I suggested some ideas in the attached. Without such explanations, the two non-timelag adjusted cospectral approached seem like artificial issues created solely for the purpose of solving them.**
- **P14L13 (from the word file): This is an impressive performance.**

  **Are there ever cases when time-lag correction is not applied first?**

  **This reads strange: would not use of time-lag-uncorrected values just be artificially making two bad cases (green and red)?**

  **Do we even need to show and discuss green and red? -- It seems that these can be removed from the manuscript.**

  **P18L5 (from the word file): I am still not sure why these two are needed to be in the paper.**

  **Running co-spectra without time lag does not seem to make sense. Of course, without time lag compensation covariances are not going to be as good.**

  **I guess I can only imagine that with very-very low fluxes, it is difficult to confidently determine the lag, so cases 1 and 2 could potentially be an illustration of the impact of inability to determine the lag.**

  **If this is indeed the case for using #1 and #2 in the manuscript, it may be good to explain it very clearly/explicitly in Introduction, and one more time in MM so reader understands the goal.**

- Regarding using time-lag uncorrected CSA approaches, it is clear that it causes confusion as those cases (i.e., $CSA_{sqrt(H)}$ and $CSA_H$) are not in use in literature. We agree with the referee. The relevant approaches are removed. The $CSA_{sqrt(H), sync}$ is the only approach used for CSA in the manuscript at the moment. We simply referred the use of square-root of the transfer function to the companion paper by Peltola et al. (2020).

**4. In the Conclusions section, it may be very useful to provide an assessment on the ease and reliability of the automaton for each of the compared techniques. I have included some ideas. The full impact of the newly proposed technique would only happen if a broad community accepts it and start using it. Automation or semi-automation is one of the keys to such acceptance and use.**

- **P19L9 (from the word file): It seems like CSA approach (fig 7 blue) can be implemented automatically without a need of the interaction from the user.**

**However, the price to pay for doing it this way is reduced performance of the correction with increased SNR, in other words, when fluxes are small.**

**PSAA20 seems to require an input from the user before correction can be properly implemented. User needs to look at the data and determine the nature of the noise, etc.**

**However, this provides a much better performance in all cases, including small fluxes (Fig 5B).**

**User interaction needed for PSA seems to be not more complex than the interactions needed for U* threshold tools. or for setting up max and min for theoretical time lag correction. So it is quite doable.**

**If the above understanding is correct, it may be worth adding a paragraph to the Conclusion regarding the ability to automate frequency corrections for small fluxes using new method or both new old and new method.**

**Perhaps, such paragraph could go here, just before "Finally, given the constraints…" and could include authors' recommendations on the automation. Maybe something similar to what I have described in this comment above, if it is correct of course.**

- It is clear that EC data processing is quite laborious, and in particular, some steps, i.e. the frequency response correction, require expertise on micrometeorology and signal processing, hence automating the process as much as possible is of great importance. We showed that the new $PSA_{A20}$ method requires less user interference compared to $PSA_{I07}$ if the type of the noise is determined in advance. However, since the limitation of the new method is not tested against different types of noise in our study, we are reluctant to provide a guideline on automating the time-constant estimation.

**P1L5 (from the word file): May be a bit confusing choice of a symbol, since H is frequently used for sensible heat flux.**

- Indeed, H is frequently used for sensible heat flux. However, in many key studies on the frequency response correction (e.g., Ibrom et al. 2007, Mammarella et al. 2009, Fratini et al. 2012), the same symbol (H) was used to describe the transfer function. Thus, we followed the literature. But still, in order to prevent the possible confusion we didn't use any symbol to describe the sensible heat flux when mentioning it in the manuscript.